# Electrophysiological and Behavioral Responses of *Batocera horsfieldi* Hope to Volatiles from *Pistacia chinensis* Bunge

**DOI:** 10.3390/insects14120911

**Published:** 2023-11-27

**Authors:** Jianting Fan, Kaiwen Zheng, Ping Xie, Yifan Dong, Yutong Gu, Jacob D. Wickham

**Affiliations:** 1National Joint Local Engineering Laboratory for High-Efficient Preparation of Biopesticide, School of Forestry and Biotechnology, Zhejiang Agriculture and Forestry University, Hangzhou 311300, China; hehaixiao2023@stu.zafu.edu.cn (K.Z.); xpfirst@zafu.edu.cn (P.X.); 2021602122119@stu.zafu.edu.cn (Y.D.); 2022602121161@stu.zafu.edu.cn (Y.G.); 2State Key Laboratory of Integrated Management of Pest Insects and Rodents, Institute of Zoology, Chinese Academy of Sciences, Beijing 100101, China; 3A.N. Severstov Institute of Ecology and Evolution, Russian Academy of Sciences, 33 Leninsky Prospect, Moscow 119071, Russia

**Keywords:** *Batocera horsfieldi*, volatiles, preference, GC-MS, EAG, *Y-tube* olfactometer

## Abstract

**Simple Summary:**

*Batocera horsfieldi* Hope (Coleoptera: Cerambycidae) has a wide host range, attacking nearly 100 tree species, and has become a devastating native forestry pest in China. Studies of various phytophagous insect attractants have shown that plant volatiles can be used as effective bait while minimizing the cost and labor of manual control. In this study, we searched for key plant-derived volatile attractants for *B. horsfieldi* by extracting and evaluating volatile components of host plants using beetle electrophysiological responses and Y-tube behavioral experiments. We provide a theoretical framework for the green control of the beetle.

**Abstract:**

Following infestation by phytophagous insects, changes in the composition and relative proportion of volatile components emitted by plants may be observed. Some phytophagous insects can accurately identify these compounds to locate suitable host plants. We investigated whether herbivore-induced plant volatiles (HIPVs) generated by herbivory on *Pistacia chinensis* Bunge (Sapindales: Aceraceae) might be semiochemicals for the host location of *Batocera horsfieldi* Hope (Coleoptera: Cerambycidae). We performed two-choice bioassays (indoor darkroom, inside cages) on plants damaged by adult feeding and intact control plants. Volatiles from these plants were then collected and identified, and the response of adult antennae to these compounds was tested via electroantennography (EAG). The behavioral responses of *B. horsfieldi* to these compounds were finally assessed using a Y-tube olfactometer. Host plant choice tests show that *B. horsfieldi* prefers feeding-damaged *P. chinensis* over healthy trees. In total, 15 compounds were collected from healthy and feeding-damaged *P. chinensis*, 10 of which were shared in both healthy and feeding-damaged *P. chinensis*, among which there were significant differences in the quantities of five terpenes, including α-pinene, β-pinene, α-phellandrene, D-limonene, and β-ocimene. In EAG assays, the antennae of *B. horsfieldi* adults responded strongly to (Z)-3-hexen-1-ol, β-ocimene, 3-carene, γ-terpinene, D-limonene, myrcene, and α-phellandrene. The antennae of *B. horsfieldi* adults responded in a dose–response manner to these compounds. Y-tube behavioral experiments showed that four compounds attracted mated females ((Z)-3-hexen-1-ol, β-ocimene, 3-carene, and α-phellandrene), two compounds ((Z)-3-hexen-1-ol and α-phellandrene) attracted males, and adults of both sexes avoided D-limonene. Feeding bioassays showed that (Z)-3-hexen-1-ol and β-ocimene could promote the feeding of *B. horsfieldi* and that D-limonene inhibited this response. These results could provide a theoretical basis for developing attractants or repellents for *B. horsfieldi*.

## 1. Introduction

*Batocera horsfieldi* Hope (Coleoptera: Cerambycidae) is a major, native woodboring forestry pest in China that feeds on nearly 100 plant species belonging to taxonomically distant plant families (Salicaceae, Juglandaceae, Fagaceae, Rosaceae, Caprifoliaceae, Betulaceae, Oleaceae, Moraceae, and Euphorbiaceae). It is distributed mainly in southwest, southern, central south, and northern China, as well as in Vietnam, Japan, India, and Myanmar [1,2,3,4,5,6,7]. Its basic biology is similar to that of other members of the subfamily Lamiinae, and its life cycle is usually completed in 2–3 years [8]. Adults emerge in the early summer and feed mainly on the branches of host plants until they are sexually mature [9].

*Batocera horsfieldi* has a sensitive olfactory system that is essential for host and mate location [10,11,12]. Olfaction is of high importance to insects. The development of specialized and sophisticated olfactory systems has enabled insects to orient and interact with their environment. The olfactory system is capable of detecting and distinguishing environmental volatiles that play a key role in behaviors such as foraging, host seeking, mating, and oviposition [13,14,15,16]. The feeding and oviposition behavior of phytophagous insects are affected by the odors released by host plants [17,18,19]. Numerous reports have shown that cerambycid beetles respond to plant volatiles. Several cerambycids, including *Asemum striatum* (Coleoptera: Cerambycidae), *Acmaeops proteus* (Coleoptera: Cerambycidae), *Xylotrechus undulatus* (Coleoptera: Cerambycidae), and *Monochamus scutellatus* (Coleoptera: Cerambycidae), are attracted to the α-pinene, β-pinene, camphene, and myrcene [20]. The volatile compounds of *Platycladus orientalis* (Pinopsida: Cupressales), such as α-thujene, α-pinene, β-caryophyllene, and nerolidol, can attract adults of *Semanotus bifasciatus* (Coleoptera: Cerambycidae) [21]. The volatile plant component 3-carene can induce a behavioral response in *Anoplophora glabripennis* (Coleoptera: Cerambycidae) and was thus verified in field trapping experiments [22].

*Pistacia chinensis* Bunge, an economically important medicinal tree, is one of the host plants of *B. horsfieldi* [23]. During initial field surveys, we found that *B. horsfieldi* preferred feeding-damaged *P. chinensis* and oriented to them. We speculate that *B. horsfieldi* uses volatile cues released by feeding-damaged *P. chinensis* for host finding and selection. Studies have shown that phytophagous insects can induce changes in the volatile component profiles of plants [24]. For example, there are large differences in terpene composition between feeding-damaged *Pinus massoniana* branches and healthy branches [25]. The southern pine coneworm moth, *Dioryctria amatella*, tends to lay eggs on 2-year-old cones damaged by *Cronartium strobilinum* [26]. In contrast, healthy ashleaf maples attract *A. glabripennis*, but their attraction to feeding-damaged ashleaf maples is significantly reduced [27].

At present, the prevention and control of *B. horsfieldi* are still limited to the application of insecticides, manual killing of larvae, and tree removal [28]. Traditional pest control of *B. horsfieldi* is rarely effective because of the crypticity of larvae feeding under the bark. Trapping with a plant-volatile-based attractant is an effective method for pest monitoring and green prevention and control, and it has been widely used in the management of many pests [29,30]. Many studies have been conducted on the biology of *B. horsfieldi* [8,31,32], but few reports have been made on the olfactory communication of *B. horsfieldi* and the screening of plant attractants. This study aimed to explore attractive compounds in host plants of *B. horsfieldi*. We analyzed volatile components in *P. chinensis* using gas chromatography–mass spectrometry (GC-MS) and measured the specific electrophysiological responses of *B. horsfieldi* antennae to a variety of chemicals found in the *P. chinensis* volatile profile using electroantennographic (EAG) analysis. Finally, feeding preference experiments were conducted to screen compounds that may have attractant activity for *B. horsfieldi*.

## 2. Materials and Methods

### 2.1. Insects and Plants

*Batocera horsfieldi* adults were collected from *P. chinensis* branches in the coastal defense forests of Yuyao, Zhejiang, China (30°13′15″ N, 121°0′41″ E). The indoor conditions were as follows: temperature 25~28 °C, relative humidity 65% ± 10%, photoperiod 14L: 10D, and feeding with *P. chinensis* planted in flowerpots as food. The beetles were maintained in insect cages, and *P. chinensis* was placed in the cages as food. After observing the mating of female insects and removing the effect of mating status, they were placed in new insect cages and raised separately for all subsequent experiments. The experiments were conducted after placing the insects and plants in an outdoor natural environment for three days to allow them to acclimatize. The insects were kept in cages in the absence of plant odors for at least 2 h before testing. Two-year-old *P. chinensis* seedlings (DBH 4 cm, tall 2 m) [23] were planted indoors in pots (21.5 cm × 23 cm) with nutrient soil (Sichuan Luyiyuan Agricultural Technology Co., Ltd., Mianyang, China). A pair of *B. horsfieldi* adults were fed for 24 h and the plants then were placed indoors for 6 h away from beetles to prevent the residues of insect pheromones on the feeding-damaged *P. chinensis* plants from affecting the experimental results.

### 2.2. Host Plant Choice Tests

The experiment was carried out in a dark room at 25~28 °C. Ten pairs of adults of *B. horsfieldi* were selected and released in the center of an insect cage (200 cm × 200 cm × 100 cm). Healthy *P. chinensis* and feeding-damaged *P. chinensis* seedlings were placed on opposite sides of the cage. After each test, we randomly changed the position of the plants to eliminate the effects of tree placement. The test time was 1 h, and it was repeated 5 times. We recorded the number of adults present on each *P. chinensis*.

### 2.3. Collection and Identification of Volatile Compounds

The volatile components of *P. chinensis* were determined using the dynamic enclosure technique coupled with atmospheric samplers (QC-1S, Beijing Municipal Institute of Labour Protection Co., Ltd., Beijing, China) [33]. In brief, oven bags (Polypropylene, 406 × 444 mm, Reynolds, Richmond, VA, USA) were cleaned with acetone and then dried at 45 °C for 30 min. Before volatile collection, the bag was carefully enclosed around the twigs of the seedlings without damaging the leaves, and the bag mouth was wrapped tightly around the stems with Parafilm (Oshkosh, WI, USA) to seal it. The enclosure was ventilated through Teflon tubing with purified air (air without volatile organic compounds (VOCs)), which was obtained by passing the air through an activated carbon filter. The enclosure was flushed with purified air for 30 min at a flow rate of 500 mL/min before collecting VOCs. We connected the Teflon tubing to the enclosure outlet, and the air containing headspace volatiles at a flow rate of 500 mL/min was passed through an adsorbent filter, consisting of 200 mg of 60–80 mesh Tenax-Ta (150 mm × 5 mm ID; Beijing Municipal Institute of Labour Protection Co., Ltd., Beijing, China) packed inside a glass pipette between two pieces of glass wool, for 6 h. The adsorbed volatiles were eluted from the cartridge with chromatography grade n-hexane (1.5 mL) with n-dodecane (100 μg/L) as the internal standard, and sample volumes were concentrated to 100 μL under nitrogen and stored in a freezer (−20 °C ) in a sealed vial. There were 5 replicates.

The samples were analyzed using an HP 7890A gas chromatograph (Hewlett-Packard, now Agilent, Santa Clara, CA, USA) fitted with an HP-5MS column (30 m × 0.25 mm ID × 0.25 μm film; J&W Scientific, Folsom, CA, USA), coupled to a 5975C mass selective detector (Agilent, Santa Clara, CA, USA). The temperature program used was 50 °C for 3 min, then increased by 5 °C/min to 200 °C, and finally increased by 8 °C/min to 280 °C. The solvent delay was 3 min. The injector temperature was 250 °C and the injections were performed in splitless mode. The electron-impact ionization mode (EI, ion energy of 70 eV, filament current 50 μA, and source temperature 325 °C) was used. Data acquisition was performed in full scan mode (MS) with a scanning range of 30–500 m/z. The chemicals used to make mass spectral and retention time matches to unknowns in the samples are shown in Table 1. A tentative identification of volatiles was made with the NIST 08 standard library and confirmed by matching the mass spectra and retention times to authentic standards [34]. We used n-dodecane as a qualitative internal standard. Compounds of each sample were eluted from the Supelco pack with 500 µL hexane containing 50 ng/µL n-dodecane. A quantitative analysis of the volatiles was carried out from the peak areas based on the response of the internal standard n-dodecane and the response factors for standard compounds.

### 2.4. Electroantennographic (EAG) Assays

An EAG (Syntech-IDAC 4, Buchenbach, Germany) was used to test the olfactory response of insects to standards [33]. Based on the results of the volatiles’ collections, and the identification of the volatiles’ experimental results, the ten volatiles with the greatest abundance or that have been previously reported to attract Cerambycidae were selected as the compounds to be tested [21,22,23]. Serial dilutions (1, 10, 100 mg/mL) of standard compounds were prepared in paraffin oil and applied (10 μL) to a filter paper strip. Solvent was allowed to stand for 45 s before the strip was placed in the Pasteur pipette. The control stimulus was paraffin oil (10 μL). Test stimulations were performed by applying puffs of air for 0.5 s through the Pasteur pipette. Puffs of the test stimuli were applied at 1 min intervals in randomized order. Puffs with paraffin oil alone were applied at the beginning and end of each trial (one antenna) and between the groups of compounds to monitor the condition of the antennal preparation. The antennae used in the experiment were all from active, apparently healthy *B. horsfieldi*. Before the test, each antenna was quickly cut off at the base and tip. Every antenna was exposed to same concentration of different compounds, and was not used for more than 25 min. The two cut ends of the antenna were immersed in electroconductivity gel on opposite conductors of an electrode fork PRG-2 EAG probe (Syntech GmbH, Steinfeld, Germany). The tip of a glass Pasteur pipette tube was inserted through a hole in a piece of glass tubing that delivered a continuous flow of purified, humidified air at 600 mL/min over the antennal preparation. The pulse stimulation air flow was 120 mL/min, and the duration of the stimulation was 0.5 s [35]. The antennal responses to test compounds to different concentrations of chemicals were recorded from at least ten *B. horsfieldi*. Voltage responses were normalized relative to the average response to paraffin oil.

### 2.5. Y-Tube Behavioral Experiments

Identifiable VOCs that increased in concentration with feeding damage were used in behavioral assays if commercially available. The responses were assessed in a glass Y-tube olfactometer (main arm 9 cm inner diameter, 40 cm length, and side arms 6 cm inner diameter, 25 cm length). The olfactometer was placed on a white laboratory table and illuminated via LED lights (Hangzhou Pheromone Biotechnology Co., Ltd., Hangzhou, China). Air was pumped (QC-1S, Beijing Municipal Institute of Labour Protection, Beijing, China) through an activated charcoal filter and humidified by passing it through a bottle with tap water before directing it into the two odor-source chambers of the olfactometer (Spherical, volume 3 L, with interfaces at both ends with an inner diameter of 45 mm and an outer diameter of 55 mm). The airflow rate was approximately 500 mL/min. Filter paper strips (20 mm × 10 mm, Newstar; Hangzhou, China) were loaded with either 10 μL of the test stimulus (e.g., 100 mg/mL *α*-Pinene in paraffin oil) or paraffin oil and placed in one of the two odorant chambers. The tests were conducted from 10:00 to 14:00. In all bioassays, the test stimulus positions were reversed after three runs to avoid any directional bias. After six replicates, the olfactometer was thoroughly washed with ethanol and rinsed with acetone before being oven-dried at 45 °C. For each bioassay, a single *B. horsfieldi* was introduced into the central arm of the Y-tube and left for 10 min to make a choice (entering the side arms connecting the odor chamber and staying for more than 30 s was counted as a choice, and failure to enter the side arms was counted as no choice). We tested 100 adult insects for each compound (female to male ratio 1:1). Before the start of the experiment, the adults were placed in a small cage in the laboratory (25~28 °C, ventilated and dry) for 2 h. The test insects were not reused.

### 2.6. Effect of Plant Volatiles on Feeding

Selected compounds that elicited behavioral responses and dilutions of standard dilutions (2 mL at 10 mg/mL paraffin oil) of selected compounds that elicited behavioral responses were evenly coated on five *P. chinensis* branches lacking foliage (30 cm long × 2 cm diameter). Treated and control (untreated) branches was placed vertically against the six corners of the hexagonal insect cage (1 m height, 1 m per side). The height of the insect cage was 1 m. We placed 6 pairs of *B. horsfieldi* adults (6 females + 6 males) in the cage. After 24 h of feeding, we measured and recorded the feeding area using parchment paper and grid paper (1 mm^2^ per grid), traced the feeding pattern, and calculated the area. The experiments were performed under laboratory conditions at 25 ± 2 °C and a photoperiod of 16 L: 8D. The test was repeated 5 times.

### 2.7. Statistical Analysis

The feeding preference data were analyzed using a chi-square test. The relative EAG response ratios of *B. horsfieldi* to compounds were determined using one-way ANOVA [36]. The behavioral response of *B. horsfieldi* to compounds was also evaluated using a chi-square test. The effects of plant volatiles on feeding were determined using one-way ANOVA. Multiple comparisons were performed via Tukey’s test and Bonferroni correction at *p* < 0.05. All statistical analyses were performed using SPSS Statistics 25 software (SPPS Inc., Chicago, IL, USA), and figures were made using Origin 2021 (OriginLab, Northampton, MA, USA). Non-responding individuals were recorded but not included in the statistical analysis [37].

## 3. Results

### 3.1. Host Plant Choice Tests

In the host plant choice tests, when *B. horsfieldi* was exposed to a healthy plant and a feeding-damaged plant, both males and females tended to choose the feeding-damaged plant (*p* < 0.05). The selection rates of male and female *B. horsfieldi* to feeding-damaged *P. chinensis* were as high as 77.42% and 76.19%, respectively (Figure 1).

### 3.2. Collection and Identification of Volatile Compounds

A total of 15 plant volatiles were collected from healthy and feeding-damaged *P. chinensis* (Table 2). β-Ocimene was the predominant compound in healthy plants and myrcene was the predominant compound in damaged plants. Seven volatile components were detected in both healthy and feeding-damaged *P. chinensis*, among which there were significant differences in five compounds, including α-pinene (*p* < 0.05), β-pinene (*p* < 0.05), and α-phellandrene (*p* < 0.05), D-limonene (*p* < 0.05), and β-ocimene (*p* < 0.05). There were three compounds (5-methyl-3-hexen-2-one, 3-carene, and camphor) in healthy *P. chinensis* that were not detected in feeding-damaged *P. chinensis*, among which 3-carene and camphor were the most abundant. Five compounds ((Z)-3-hexen-1-ol, myrcene, 3-thujene, γ-terpinene and (1S,3R) -(Z)-4-carene) from feeding-damaged *P. chinensis* were not detected in healthy *P. chinensis*. Myrcene, γ-terpinene, and (Z)-3-hexen-1-ol were present at the highest concentrations.

### 3.3. Electroantennographic (EAG) Assays

The relative EAG response ratios of *B. horsfieldi* males and females to 10 compounds at three concentrations (1 mg/mL, 10 mg/mL, 100 mg/mL) are shown in Figure 2. At the 1 mg/mL concentration, all tested compounds elicited EAG responses from the beetles, with the relative response ratios of EAG ranging from 0.89 ± 0.07 for D-limonene to 2.64 ± 0.60 for (Z)-3-hexen-1-ol in males, and from 0.94 ± 0.10 for D-limonene to 2.03 ± 0.39 for α-phellandrene in females (Figure 2a). The EAG response values of females at a concentration of 1 mg/mL did not differ among the compounds (*p* > 0.05), expect myrcene. But the relative EAG response value of (Z)-3-hexen-1-ol and β-ocimene at a dose of 1 mg/mL in male adults was significantly different from that with the other compounds (Figure 2a). At the 10 mg/mL concentration, the maximum relative EAG response value of (Z)-3-hexen-1-ol produced by the males of *B. horsfieldi* was 4.78 ± 1.39, which was higher than the responses to the other compounds (Figure 2b). The females of *B. horsfieldi* also produced the greatest EAG response to (Z)-3-hexen-1-ol at a concentration of 10 mg/mL, which was significantly higher than that of other compounds (*p* < 0.05), except for β-ocimene and α-phellandrene (Figure 2b). At the 100 mg/mL dose, the relative EAG reaction value of male *B. horsfieldi* to (Z)-3-hexen-1-ol was also the highest, which was 6.96 ± 0.41, significantly higher than that of γ-terpinene, myrcene, α-pinene, β-pinene, camphor, and α-phellandrene (Figure 2c). At the 100 mg/mL concentration, the relative EAG response value of female *B. horsfieldi* to α-phellandrene was the highest, followed by (Z)-3-hexen-1-ol, and D-limonene, and the weakest stimulants were α-pinene, β-pinene, and camphor for both sexes (Figure 2c). Meanwhile, there were no significant differences between male and female adults to the same compound (*p* > 0.05).

### 3.4. Y-Tube Behavioral Experiments

Seven host plant volatiles, including (Z)-3-hexen-1-ol, β-ocimene, 3-carene, γ-terpinene, D-limonene, myrcene, and α-phellandrene, which elicited strong EAG responses with *B. horsfieldi* adults, were selected for behavioral experiments. We found that the total number of female insects selecting (Z)-3-hexen-1-ol (df = 1, *p* < 0.01), β-ocimene (df = 1, *p* < 0.01), 3-carene (df = 1, *p* < 0.01) and α-phellandrene (df = 1, *p* < 0.01) was significantly higher than that in the control group (Figure 3a). The total number of male insects selecting (Z)-3-hexen-1-ol (df = 1, *p* < 0.01) and α-phellandrene (df = 1, *p* < 0.01) was significantly higher than that selecting the paraffin oil (Figure 3b). The selection rate of D-limonene by both *B. horsfieldi* females and males was significantly lower than that of the control group (*p* < 0.05), with only 11.11% (SE = 4.65) and 11.30% (SE = 4.79).

### 3.5. Effect of Plant Volatiles on Feeding

The amount of bark removed by feeding *B. horsfieldi* with *P. chinensis* branches coated with (Z)-3-hexen-1-ol and β-ocimene was significantly higher than that of the control and other compounds (*df* = 5, *p* < 0.05, n = 30). The amount of bark removed by feeding on *P. chinensis* coated with D-limonene was the lowest. There was no significant difference in the amount of bark removed by feeding *B. horsfieldi* on *P. chinensis* coated with 3-carene and α-phellandrene compared with the control (*p* > 0.05) (Figure 4).

## 4. Discussion

This report showed that the adults of *B. horsfieldi* prefer feeding-damaged *P. chinensis* compared with healthy hosts, which was consistent with field observations. In the choice experiment, female and male *B. horsfieldi* selected damaged *P. chinensis* three times more often than undamaged plants. Sixteen VOCs were extracted and identified using air sampling and GC-MS from *P. chinensis* in healthy and feeding-damaged plants. Seven identical compounds were found in the volatile components of *P. chinensis* in two physiological states. Notably, five compounds (α-pinene, β-pinene, α-phellandrene, β-ocimene, and D-limonene) were significantly higher in damaged plants than in controls. Some plants respond to phytophagous insect attacks by producing a mixture of metabolites with varying amounts or proportions [37]. After *P. chinensis* was used to feed *B. horsfieldi*, the contents of α-pinene, β-pinene, α-phellandrene, and β-ocimene in *P. chinensis* increased significantly, whereas the contents of D-limonene decreased significantly. When plants are harmed by phytophagous insects, they release volatiles that are different from those in healthy plants [18,38]. The feeding behavior of *B. horsfieldi* also causes the host plant to release different VOCs compared to the healthy plant [6]. Six volatile components not detected in healthy *P. chinensis* were detected in *P. chinensis* after feeding by *B. horsfieldi*, among which (Z)-3-hexen-1-ol, myrcene, and γ-terpinene were detected in the highest concentrations. Three volatile components were only detected in healthy *P. chinensis*, among which 3-carene was the most abundant. The ability of these ten compounds to influence other phytophagous insects has been reported [18,39,40,41,42,43,44,45,46], and they were considered putative VOCs as attractants of *B. horsfieldi*.

Volatiles produced by plants can attract or repel insects [47,48] and cause strong EAG responses and attraction of cerambycids. The plant volatile α-pinene can attract *M. alternatus* [49], *Anoplophora nobilis* [50], *Spondylis buprestoides* and *A. striatum* [40]. Ovipositing female *M. alternatus* prefers stressed *Pinus massoniana* over healthy trees, α-pinene, β-pinene and D-limonene are more abundant in stressed trees than in healthy ones, and α-pinene has a good attraction to *M. alternatus* [51]. Female *Arhopalus tristis* strongly prefers the volatiles of burnt vs. unburnt pine bark because monoterpenes have been isolated from the volatiles of smoldering wood and monoterpenes attracted female *A. tristis* [32]. Phytosterols and alkanes, the volatile components of *Viburnum awabuki*, can cause a strong EAG response in *B. lineolata* [6]. The results showed that all ten volatile components could cause EAG responses in *B. horsfieldi*, and there were significant differences among them. The EAG response amplitude of *B. horsfieldi* to different concentrations of (Z)-3-hexen-1-ol was particularly strong, and the response value of each compound increases gradually with increasing concentration, but there was no significant difference in the EAG responses of male and mated female insects to the same compound in our study. α-Pinene, which is attractive to other cerambycids, did not cause the strong EAG response in *B. horsfieldi*, and its relative EAG response value was lowest among all compounds, most likely because *P. chinensis* is a deciduous tree, which has profiles of volatiles that are not dominated by α-pinene [52,53].

Insect-induced volatiles can be easily detected by insects, and phytophagous insects can decipher the state of host plants and make corresponding behavioral responses through the information contained in volatiles [54,55]. Seven compounds with strong EAG responses were selected for Y-tube behavioral experiments, including two volatile components produced by the feeding behavior of *B. horsfieldi*. The results showed that four compounds ((Z)-3-hexen-1-ol, β-ocimene, 3-carene, and α-phellandrene) had obvious attractant activity to mated female insects, while two compounds ((Z)-3-hexen-1-ol and α-phellandrene) had attractant activity to male insects, and D-limonene had obvious repellant effects to adults of both sexes. (Z)-3-hexen-1-ol was reported to have a good attraction to *Empoasca onukii* and some cerambycids [39,56]. β-Ocimene was found to attract female *Aphidius ervi* and parasitoids. Also, β-ocimene has been shown to increase both mating and oviposition rates in *Hyphantria cunea* [57]. The bicyclic monoterpene 3-carene has been reported to be an attractant for *Dendroctonus* and *Hylurgops* [58]. α-Phellandrene can cause cuticular damage to *Lucilia cuprina* larvae [59]. D-Limonene has antibacterial and antiviral activities [60,61]. These compounds may play an important role in nature and influence the behavior of *B. horsfieldi* at different release rates. Among cerambycids, sexually dimorphic behavioral responses have been reported in *B. horsfieldi* and reflect differences in the role played by the same VOC in the ecology of males and females [62]. In the field, volatiles do not exist alone, and insects therefore rely on multiple compounds to locate host plants [63].

Our results show that the volatiles (Z)-3-hexen-1-ol and β-ocimene from host plants in indoor environments can attract and significantly improve the feeding intensity of *B. horsfieldi* on host plants, and that D-limonene has an obvious repellant effect on *B. horsfieldi*. However, the attraction effect of plant volatiles to *B. horsfield* in the forest remains uncertain [9]. If different compounds are mixed in a certain proportion, their attractiveness may be enhanced [63]. Future studies are necessary to further explore the effective attractant components of *B. horsfieldi* and their effect on *B. horsfieldi* alone or in combination, and this should provide a theoretical framework for the development of attractants or repellents for pest management.

## Figures and Tables

**Figure 1 insects-14-00911-f001:**
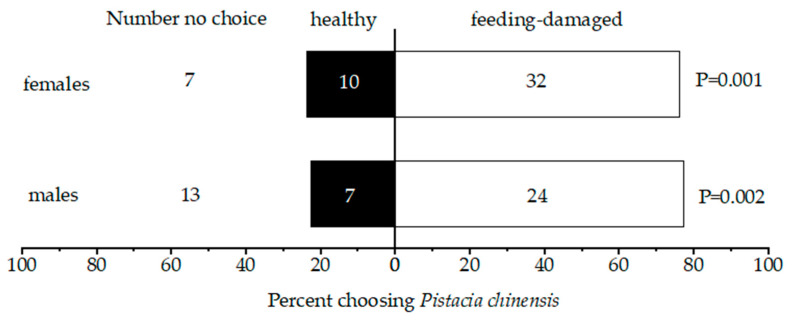
Results of selection for healthy *P. chinensis* and feeding-damaged *P. chinensis* by *Batocera horsfieldi*. The numbers in bars represent the total number of *B. horsfieldi* individuals who chose one or the other plant treatment.

**Figure 2 insects-14-00911-f002:**
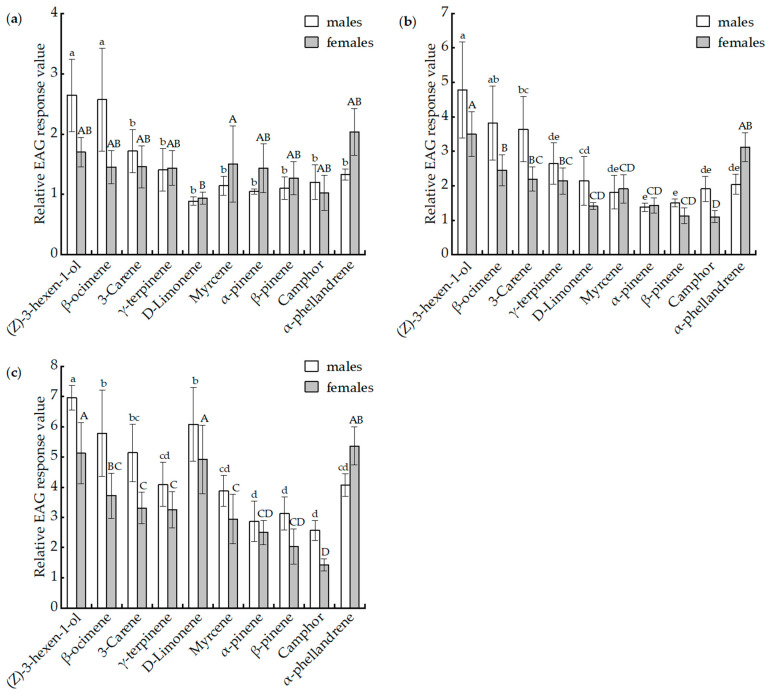
EAG responses of *Batacera horsfieldi* females and males to different doses of ten components in the volatile of *Pistacia chinensis.* (**a**) 1 mg/mL, (**b**) 10 mg/mL, and (**c**) 100 mg/mL. Different lowercase and capital letters above bars indicate significant differences (Tukey’s test with Bonferroni correction, *p* < 0.05), in the EAG responses of male and female adults, respectively, to different volatile compounds.

**Figure 3 insects-14-00911-f003:**
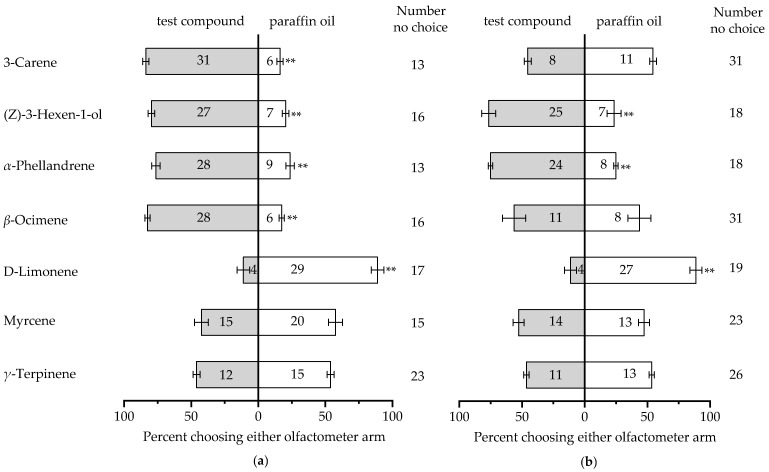
Response in a Y-tube olfactometer by *Batocera horsfieldi* females (**a**) and males (**b**) to ten compounds. The numbers in bars represent the total number of *B. horsfieldi* that chose the olfactometer arm. Asterisks indicate significant differences (** *p* < 0.01, chi-square test).

**Figure 4 insects-14-00911-f004:**
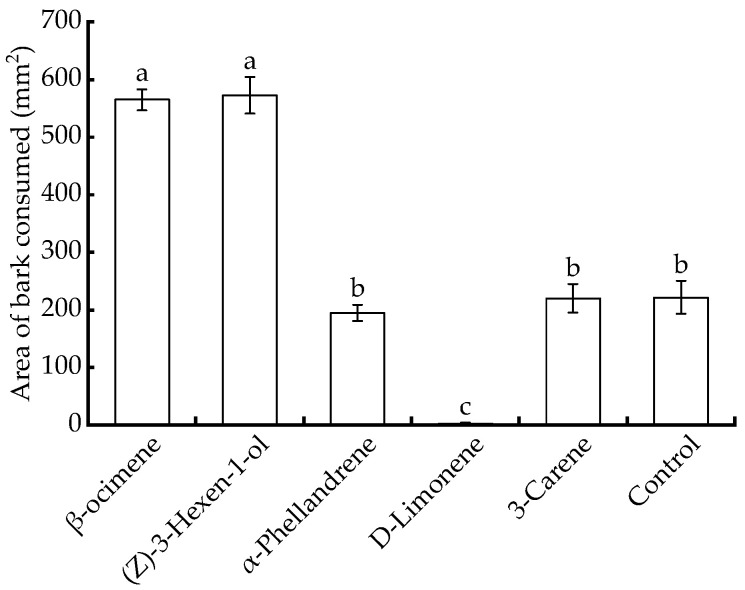
Effect of plant volatiles on the feeding of *Batocera horsfieldi*. Different lower-case letters indicate significant differences in the bark area consumed by *B. horsfieldi* on *Pistacia chinensis* branches coated with different volatile substances (one-way ANOVA, Tukey’s test with Bonferroni correction. *p* < 0.05).

**Table 1 insects-14-00911-t001:** Purity and source of chemicals used in the experiments.

Chemicals	Purity%	Compound Source
Ethanol	Analytically pure 95%	Macklin Biochemical Technology, Shanghai, China
Acetone	Analytically pure 99.5%	J&K Scientific, Shanghai, China
n-Hexane	Analytically pure 95%	J&K Scientific, Shanghai, China
Paraffin oil	99%	Hushi Laboratorial Equipment Co., Ltd., Shanghai, China
α-Pinene	99%	J&K Scientific, Shanghai, China
β-Pinene	98%	J&K Scientific, Shanghai, China
γ-Terpinene	95%	J&K Scientific, Shanghai, China
β-Ocimene	98%	J&K Scientific, Shanghai, China
(Z)-3-Hexen-1-ol	98%	J&K Scientific, Shanghai, China
Myrcene	95%	J&K Scientific, Shanghai, China
n-Dodecane	99%	J&K Scientific, Shanghai, China
n-Hexane	99%	J&K Scientific, Shanghai, China
Camphor	98%	Tokyo Chemical Industry Co., Ltd., Tokyo, Japan
D-limonene	95%	Tokyo Chemical Industry Co., Ltd., Tokyo, Japan
3-Carene	90%	Acros Organics, Morris, NJ, USA
α-Phellandrene	85%	Sigma-Aldrich, St. Louis, MO, USA

**Table 2 insects-14-00911-t002:** Volatile emission of compounds from *Pistacia chinensis* in different states in μg/L ± SE.

No.	Compounds	CAS no.	Retention Time (min)	Extraction Solution Concentration (μg /L)	
				Healthy	Feeding-Damaged
1	(Z)-3-Hexen-1-ol	928-96-1	7.398	-	6.74 ± 2.46
2	Styrene	100-42-5	7.630	38.51 ± 7.47	41.17 ± 11.56
3	α-Pinene	80-56-8	8.424	10.39 ± 1.49	20.83 ± 4.04 *
4	5-Methyl-3-hexen-2-one	5166-53-0	9.105	1.39 ± 0.30	-
5	β-Pinene	18172-67-3	9.402	7.43 ± 1.76	15.12 ± 1.56 *
6	Myrcene	123-35-3	9.996	-	89.23 ± 19.20
7	3-Thujene	2867-5-2	10.082	-	0.23 ± 0.21
8	α-Phellandrene	99-83-2	10.110	1.61 ± 1.02	5.85 ± 1.35 *
9	2-Carene	554-61-0	10.736	4.95 ± 1.34	5.28 ± 1.80
10	γ-Terpinene	99-85-4	10.801	-	20.36 ± 4.37
11	3-Carene	13466-78-9	10.288	8.9 ± 2.32	-
12	D-limonene	5989-27-5	10.779	25.04 ± 5.80	9.14 ± 2.25 *
13	(1S,3R)-(Z)-4-Carene	5208-49-1	10.882	-	6.33 ± 1.33
14	β-Ocimene	3779-61-1	10.968	133.07 ± 24.13	66.58 ± 12.36 *
15	Camphor	464-49-3	13.902	6.30 ± 1.32	-

The dashed line (-) indicates that the compound was not detected in the volatiles. * indicates significant differences in the content of *Pistacia chinensis* in different states (chi-square test, *p* < 0.05).

## Data Availability

The data supporting the findings of this study are available from the corresponding author upon reasonable request.

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
