# Peer review of "Electrophysiological and Behavioral Responses of Batocera horsfieldi Hope to Volatiles from Pistacia chinensis Bunge"

_insects, 2023, doi:10.3390/insects14120911_

Round 1

Reviewer 1 Report

Comments and Suggestions for Authors

The manuscript describes some well-conceived and executed experiments to identify semiochemical host-location cues for a cerambycid beetle, specifically, semiochemicals produced when conspecifics feed on a host plant. The findings make a significant contribution to the science, have practical value, and should be of interest to the readership of Insects. The methods are generally sound and appropriate for addressing the hypotheses of the scientists.  However, the paper has some significant weaknesses that will require revision and additional review.  The study was not sufficiently controlled for concluding that the compounds identified post-feeding were HIPVs since there was no sampling of mechanically-damaged plants. Any damage to the plant could result in an increase in release of volatiles, and there may be nothing specific to beetle feeding that caused the appearance or increase of the compounds that were observed.  HIPVs are only a possibility and must be discussed as such. The statistical analyses do not seem appropriate for some tests, and I would like the authors to seek assistance from a statistician in revising their analyses.  Specific problems with the statistics are indicated in the line-by-line comments below.  Also, there appear to be numerous solvent contaminants (ones commonly present in their extraction solvent, hexane) included in their list of compounds detected from damaged and undamaged plants (Table 2).  The authors need to carefully check whether these are contaminants and, if they are, remove them from the table and their reporting/discussion of the data.  The discussion section is largely composed of repetition of results with minimal discussion of their ecological and practical significance or their relationship to existing research on the chemical ecology of cerambycid beetles.  The English expression and grammar is typically insufficient for publication (although normally the text is fully understandable), and I have made extensive efforts to help the authors with this.  English grammar, expression, and clarity should be checked again by reviewers or the editor before the article is accepted.

L13  Indicate insect range, and whether it is native or invasive.  Insert “(Coleoptera: Cerambycidae)” after species name

L15 Minimizing cost and labor of doing what?

L16 Change “components” to “attractants for B. horsfieldi”; change “identifying” to “evaluating”

L17 Delete “olfactory”.  Remove italics from “Y-tube”.  Do this throughout the paper.

L19 Change “As a result of their” to “Following”

L20 Change “can be” to “may”

L21 Change “aimed to investigate” to “investigated”

L21-23 Change sentence to: “We investigated whether herbivore-induced plant volatiles (HIPVs) generated by herbivory on Pistacia chinensis Bunge might be semiochemicals for Batocera horsfieldi Hope.” Indicate order/family for species here and throughout paper whenever a species is named for the first time.  Indicate in a few words the importance of these species.

L24 Indicate bioassay type (laboratory, greenhouse, olfactometer type, etc.).

L25 “Electroantennography” shouldn’t be capitalized. Change “on” to “to”

L26 Change “selectivity” to “behavioral responses”.  Change “for” to “to”

L27-28 “Perching” is not feeding; a feeding bioassay measures amount of feeding and not other types of behaviors.

L29 Delete “plants of”

L301-31 Replace “compounds…terpenes” with just “terpenes”

L34 Delete “in EAG studies”

L36 Delete “obvious”.  Change “to” to “with”

L37-38 Change “had obvious avoidance reaction to” to “avoided”

L38 Change “The test results…feeding” to “Feeding bioassays”.  Delete “the”

L39 Delete comma.

L40 Change “ the feeding intensity of horsfieldi” to “it.”

L53 “had” to “has”

L55 “orientate” to “orient”

L56 Delete “thousands of”

L58-60 Delete this sentence.  The content is largely inaccurate, and it is simpler to remove it than change it.

L61 Delete “the volatile” and change “odor” to “odors”

L61-62  Change to: “Numerous reports have shown that cerambycid beetles respond to plant volatiles.”

L62,  Change “Many” to “Several”   Species names need authorities.

L67 Change “the” to “an”   Presumably a behavioral (not olfactory per se) response was demonstrated with the trapping experiment.

L69-70 Delete sentence.  Results cannot be reported in the introduction.  Is there some other basis for making your hypothesis, or can this observation of “perching” (if it is the reason you examined HIPVs) be removed from the results and just be here?

L69 Is Pistacia chinensis an economically/ecologically important species?  Why?

L71 Utilizes to do what?

L72-73  Delete sentence.  This idea was expressed in the previous paragraph.

L74 Change “There” to “For example, there”.   Change “variations” to “differences”.

L75 Delete “and their relative contents”.

L79 Delete “and even had a repellant effect”.   Change “was” to “is”

L79-81 Delete sentence.  Not sufficiently relevant.

L82 Delete “methods”. 

L83 If .“ application of insecticides and tree removal” can correctly replace “chemical and physical control”, make this change (i.e., be more specific).

L86-87 Change to “Many studies have been conducted on the biology of B. horsfieldi [8,38,39].”

L88 Delete “the” and “mechanism”.

L90-91 Do not capitalize “gas chromatography-mass spectrometry”.

L91 Comma after “(GC-MS)”.

L93 Delete both instances of “the”.

L100 Delete text in parentheses.

L101 Replace “Branch feeding was”  With “Beetles were”.

L102 “planted” = in pots?

L101-102 Describe temperatures and lighting.

L103 Were they mated to identify the sex or for some other purpose? 

L104 “and raised”?  I don’t know what this means.  Instead: “used”?

L104-105 “adaptation”?  Do you mean “acclimatization”?  How were the plants “adapted”? 

L105 Where were they kept? 

L106 How was it determined that the female was mated? L103 indicates you already knew they were mated.

L108 What is “nutritional soil”?  Gravel or sand? Gravel is typically centimeters in diameter.

L109 Delete “as healthy P. chinensis”. 

L109-110 I don’t understand “then placed for 6 h”.   What was placed?  Placed where?  The feeding damage was produced during 24 or 6 hours?  Was the amount of feeding assessed in some way?  The amount of feeding damage could influence the plant’s response. Delete text in parentheses.

L112 What was your source of lighting?

L115 Was the position of the treatments in the cage randomized (or at least alternated between sides of the cage for each trial)?  

L116 “perching” is an odd word to use for an insect behavior (it is typically used just with birds).  I can’t think of a good replacement, however.  Use “present”.

L118-120 Please provide literature citation(s),

L120 Indicate the material of the oven bags (I think it is polyester).

L121 How long were they dried?

L121-122  How was the bag sealed (e.g., wrapped tightly around the stem)?

L123-124 Activated charcoal will only remove VOCs.  Merely state that the air was purified by passing through an activated charcoal filter. Replace “zero” with “purified” or “filtered”.

L125 500 µl/min is a miniscule and vastly insufficient amount of air.  Do you mean ml/min?

L126-127 What do you mean by “constant volume of air”?  How was mixing obtained?

L129 I doubt that BMILP manufactured the Tenax cartridges.  Provide the name and address of the manufacturer.

L130 What was the air flow rate through the cartridges?  If you know the temperature at which the aerations were performed, indicate this. Was there a test for breakthrough through the cartridges?

L131 Use “cartridge” not “collector”.   Unless there was some specific reason for doing three sequential rinses with the same solvent, simply indicate that the cartridges were eluted with 1.5 ml hexane.  Delete “solvent”.

L133 Delete comma and “at”.

L137 Delete “the”.

L140 Delete “set to”.

L143 Used for what?  I assume for making mass spectral and retention time matches to unknowns in the samples, but indicate this. Explain the quantitation method used.  Did you calculate a response curve for each compound?  Did your identifications have both a mass spectral and retention time match with your standards?

L147 How were compounds chosen for the EAG tests?

L148 Delete “antennal potential measurement system”

L149 Delete “the”.  Please provide a citation for the methods used.

L150 “Paraffin” should not be capitalized.

L151-152 Paraffin oil is not volatile and thus did not evaporate.

L153 Change “a” to “the”. 

L154 Delete “the filter paper with the stimuli”.

L155 For how long was a single antennal preparation used? 

L156 Replace “each experiment” with “trial (one antenna)” if this is correct.

L158 Delete “with good activity” and change to “from active, apparently healthy B. horsfieldi”

L160 What kind of electrodes?  An electrode fork?  If so indicate the model and manufacturer.  “Air flow rate”?  I assume this is the air flow delivering the odor?  How was the odor delivered (in glass tubing, etc.)?

L163 Replace “EAG” with “voltage”.  Delete “with respect to the control”.

L165 Averaged across the entire trial?

L167  I think you mean to say “Identifiable VOCs that increased in concentration with feeding damage were used in behavioral assays if commercially available.”

L171 Were the lower surfaces inside the Y-tube lined, or was there simply a white surface beneath the Y-tube?

L172 Indicate source of light.

L173  Simply “humidified”.

L174 What did the odor source chambers consist of?

L177 Insert paraffin oil” after “mL”

L182 What did a “choice” consist of?  Presence of the insect in one of the two branches, contact with the stimulus, or something else?  Were the insects prevented from touching the filter paper and how was this accomplished?  How many insects of each sex were tested with each stimulus?  Were insects re-used? Under what conditions were the insects kept prior to bioassays?

L184 Change to “Dilutions of standard compounds (2 mL at 10 mg/ml) were evenly coated...”.   How were the compounds selected for testing? 

L185 How were the branches presented?  Were they simply lying horizontally on the floor or were they supported in some way?  Did the branches include foliage?

L187 Write: “…hexaganol insect cage (1 m height, _ m per side)…” and delete next sentence.  Indicate what was your control in the first sentence of the paragraph.

L188 Replace “insect” with “the”.  I don’t understand “the feeding notch on the branch was recorded”.   It might be sufficient to indicate that the surface area of feeding damage was recorded. 

L190 How was the area measured on the graph paper?

L193-194 What were these tests used to compare (what was the dependent variable)?  The LSD test presumably was used with an ANOVA; please explain this.  Please indicate whether a correction was made to the LSD test for experiment-wise error (i.e., a Bonferroni correction).  Normally, an LSD test is not considered acceptable without such a correction. Same for L195-196.  How was a chi-square test – used for count data – used in analysis of the EAG data?

L193-196 How were tests done for the Y-tube assays and the quantities of plant volatiles?

L202 Use “host plant choice tests” instead of “perching selectivity experiments”

L204 If a t-test was performed, then it requires a t-distribution, not an F-distribution.

L208 Change “perching selectivity” to “host preference tests”.  Change “to” to “with”

L209-210.  Change “P. chinensis” to “one or the other plant treatment”.

L212 Delete “number”.

L213-217 This information should go to the methods section.  If the mass spectra of all compounds were matched to spectra of standards, you don’t need to mention use of the NIST library.  Compounds identified with the NIST library alone should be indicated as tentative.

L214 Delete “, the components…computer,”

L216-217 Delete “detected…classes:”

L217 Benzene is a specific compound; the class is “aromatics”.  These compounds (most of those listed as “benzene”) are common contaminants of hexane.  If these (and other compounds) were in your extraction solvent (which is very likely) they should be removed from the table.  An extract of an unused cartridge should have been used as a control to eliminate these compounds from the data.  I strongly suspect that toluene, all three xylenes, ethylbenzene, and styrene, as well as several other compounds in the list, are solvent contaminants.

L218  Make reference to table 2.

L218-219  Condense these two sentences to “β-ocimene was the predominant compound in healthy plants and myrcene in damaged plants.”  

Table 2:  Units of “Content” need clarification.  Per liter extraction solvent or air drawn into the cartridges?  It is more typical for this kind of data to be reported as amount per sample.

L221 Delete “identical”.

L227 Change “which have not been” to “that were not”.

L231 Delete comma and “the”.

L232 Change “highest” to “present in highest concentrations”.

L233 Many compounds listed as terpenes are not. Terpenes are unsaturated compounds with carbons in multiples of five (5,10,15 etc.)

L234-235  An LSD test is used for multiple comparisons.  Only two values are being compared here, it seems.  A t-test is appropriate.  With just a single contrast, indicate a difference with an asterisk rather than significance lettering.  Change “in different states” to “of the two treatments”.

L239-241

L242 What is meant by “measurable”? 

L257-260 If differences were not statistically significant, they should not be mentioned. Also, if there were no differences between males and females, male and female data should be pooled and data presented and discussed for sexes combined.

L264 “differences”

L265 Insert “, respectively,” after “female”

L265-267 Delete this sentence since (according to the figure) there were no differences between males and females for any single compound.

Figure 2.  There appear to be no differences between males and females.  If this is true, remove the “ns” markers.  It would also be helpful to pool data from males and females if sexes did not differ.  The graph would be easier to interpret, and you might detect more significant differences among compounds. 

L270 Change “which…value of” to “which elicited the strongest EAG responses from”

L271-272 Change this sentence to “Selection of the treatment over the control olfactometer arm by females was greatest for (Z)-3-hexen-1-ol (80% ± 3% SE), β-ocimene (83% ± 2% SE), …” etc. for this and other such sentences in the paragraph. The percentages should have only two significant figures.

L274-277 It does not seem the stats were appropriate here.  A t-test was not appropriate if percentage responses to the two arms were compared, since these values are not independent (the value of one is dependent on the value of the other).  Also, the data are counts of responses, and a t-test is not appropriate for these kinds of data. The correct test for 2-choice olfactometers used with single insects is a G-test or exact probabilities test where the proportions of insects responding to each arm (e.g., 2 vs. 8) is contrasted with the null hypothesis of no preference (5:5)

L275 Change “selected for” to “selecting”

L276 Change “were” to “was”.

L285 Change “Olfactory behavior of” to “Response in a Y-tube olfactometer by”

Figure 3.  Delete “<---> paraffin oil” and “odor source combinations” from the left-hand titles. Instead, write “test compound” above the left-hand bars and “paraffin oil” above the right-hand bars at the top of each plot

L290-292  This information should only be given in the methods section.   An explanation of why these specific compounds were chosen should be provided.

L292 Change “feeding area” to “amount of bark removed by feeding”

L293 Delete “, which…mm2,” Data are sufficiently presented in the graph and don’t need repetition here.  Replace first comma with “and”.

L295-298 See comment for line 293.

L300 “lowercase” to “lower case”

L301 Change “significantly” to “significant”.  Change “feeding intensity of” to “bark area consumed by”.  Change “to” to “on”.

Figure 4:  On Y-axis label, change “Feeding area” to “Area of bark consumed”

L304 Delete “to perch on”.

L305 Provide a citation of these previous observations, if published.

L306 Replace “the selection…plants” with “female and male B. horsfieldi selected damaged P. chinensis 3-fold more often than undamaged plants.”

L304-371 The discussion section is mostly repetition of results and repetition of information already presented sufficiently in the introduction.  The data is very interesting and deserves sufficient commentary so the reader can appreciate this. You leave many important topics unaddressed. For example, do your results change our understanding of HIPVs?  Have similar results been found for other cerambycids? How are they similar and how different?  Are all or some of the findings novel and in what way? Why might this be?  Have these same compounds been found to influence beetle behavior in other cerambycid-host systems? What was concluded about their ecological role in these systems?  What is known about the compounds that produced responses (are they common or rare in plants, and how might this impact behaviors in nature)? Why would some be repellant and others attractive (e.g., why would limonene be special)– both generally and in the specific case of B. horsfieldi?  How could your findings be used in applications?  Are there specific problems or needs in pest management that could be addressed for B. horsfieldi with your findings?  Should HIPVs such as you found mediate beetle aggregation on damaged hosts and influence spatial distribution of damage? You don’t need to answer all of these questions, but addressing those you feel most important will greatly improve the value of your work to readers. Your hard and careful work in the laboratory deserves such explanations.  Only repeat the reporting of results when necessary to support your answers to the questions above. 

  Comments on the Quality of English Language

There are 2-3 sentences that cannot be understood, but generally I believe I had no difficulty in understanding the authors' intended meaning.  I did not feel that the language limited my ability to do a thorough review.  Most sentences had at least one issue with grammar or syntax, and -- except for the discussion which requires substantial re-writing -- I identified the issues and recommended corrections within my comments to the authors.  Use of English vocabulary was generally quite good.

Author Response

Point 1: L13 Indicate insect range, and whether it is native or invasive.  Insert “(Coleoptera: Cerambycidae)” after species name

Response 1: Thank you for your correction, we have amended this here and also the first sentence of the Introduction.

Point 2: L15 Minimizing cost and labor of doing what?

Response 2: This is to minimize the cost and labor of manual controls, we have amended this.

Point 3: L16 Change “components” to “attractants for B. horsfieldi”; change “identifying” to “evaluating”

Response 3: Done

Point 4: L17 Delete “olfactory”.  Remove italics from “Y-tube”.  Do this throughout the paper.

Response 4: Done

Point 5: L19 Change “As a result of their” to “Following”

Response 5: Done

Point 6: L20 Change “can be” to “may”

Response 6: Done

Point 7: L21 Change “aimed to investigate” to “investigated”

Response 7: Done

Point 8: L21-23 Change sentence to: “We investigated whether herbivore-induced plant volatiles (HIPVs) generated by herbivory on Pistacia chinensis Bunge might be semiochemicals for Batocera horsfieldi Hope.” Indicate order/family for species here and throughout paper whenever a species is named for the first time.  Indicate in a few words the importance of these species.

Response 8: Thank you for your correction, we have corrected this.

Point 9: L24 Indicate bioassay type (laboratory, greenhouse, olfactometer type, etc.).

Response 9: We did indoor darkroom bioassays inside cages We have indicated this.

Point 10: L25 “Electroantennography” shouldn’t be capitalized. Change “on” to “to”

Response 10: Done

Point 11: L26 Change “selectivity” to “behavioral responses”.  Change “for” to “to”

Response 11: Done

Point 12: L27-28 “Perching” is not feeding; a feeding bioassay measures amount of feeding and not other types of behaviors.

Response 12: Thank you for your suggestion, we have corrected this to indicate beetles chose feeding damaged plants.

Point 13: L29 Delete “plants of”

Response 13: Done

Point 14: L301-31 Replace “compounds…terpenes” with just “terpenes”

Response 14: Done

Point 15: L34 Delete “in EAG studies”

Response 15: Done

Point 16: L36 Delete “obvious”. Change “to” to “with”

Response 16:  Revised

Point 17: L37-38 Change “had obvious avoidance reaction to” to “avoided”

Response 17: Done

Point 18: L38 Change “The test results…feeding” to “Feeding bioassays”. Delete “the”

Response 18: Done

Point 19: L39 Delete comma.

Response 19: Done

Point 20: L40 Change “ the feeding intensity of horsfieldi” to “it.”

Response 20: Done

Point 21: L53 “had” to “has”

Response 21: Done

Point 22: L55 “orientate” to “orient”

Response 22: Done

Point 23: L56 Delete “thousands of”

Response 23: Done

Point 24: L58-60 Delete this sentence. The content is largely inaccurate, and it is simpler to remove it than change it.

Response 24: Done

Point 25: L61 Delete “the volatile” and change “odor” to “odors”

Response 25: Done

Point 26: L61-62  Change to: “Numerous reports have shown that cerambycid beetles respond to plant volatiles.”

Response 26: Done

Point 27: L62,  Change “Many” to “Several”   Species names need authorities.

Response 27: Done

Point 28: L67 Change “the” to “an”   Presumably a behavioral (not olfactory per se) response was demonstrated with the trapping experiment.

Response 28: Thank you for your correction, we have corrected this.

Point 29: L69-70 Delete sentence.  Results cannot be reported in the introduction.  Is there some other basis for making your hypothesis, or can this observation of “perching” (if it is the reason you examined HIPVs) be removed from the results and just be here?

Response 29: Thank you for your correction, we have corrected this. We found the phenomenon during exploration of the open field and next observed the beetles perching on feeding damaged plants.

Point 30: L69 Is Pistacia chinensis an economically/ecologically important species?  Why?

Response 30: Yes, it is. Pistacia chinensis is a medicinal plant and the recently discovered host plant of Batocera horsfieldi Hope in Yuyao. We noted this in the manuscript.

Point 31: L71 Utilizes to do what?

Response 31: Ulitizes host volatile cues.

Point 32: L72-73  Delete sentence.  This idea was expressed in the previous paragraph.

Response 32: Done

Point 33: L74 Change “There” to “For example, there”.   Change “variations” to “differences”.

Response 33: Done

Point 34: L75 Delete “and their relative contents”.

Response 34: Done

Point 35: L79 Delete “and even had a repellant effect”.   Change “was” to “is”

Response 35: Done

Point 36: L79-81 Delete sentence.  Not sufficiently relevant.

Response 36: Done

Point 37: L82 Delete “methods”.

Response 37: Done

Point 38: L83 If .“ application of insecticides and tree removal” can correctly replace “chemical and physical control”, make this change (i.e., be more specific).

Response 38: Thank you for your correction, we have corrected this.

Point 39: L86-87 Change to “Many studies have been conducted on the biology of B. horsfieldi [8,38,39].”

Response 39: Done

Point 40: L88 Delete “the” and “mechanism”.

Response 40: Done

Point 41: L90-91 Do not capitalize “gas chromatography-mass spectrometry”.

Response 41: Done

Point 42: L91 Comma after “(GC-MS)”.

Response 42: Done

Point 43: L93 Delete both instances of “the”.

Response 43: Done

Point 44: L100 Delete text in parentheses.

Response 44: Done 

Point 45: L101 Replace “Branch feeding was”  With “Beetles were”.

Response 45: Done

Point 46: L102 “planted” = in pots?

Response 46: Yes, planted in pots. Corrected.

Point 47: L101-102 Describe temperatures and lighting.

Response 47: Done

Point 48: L103 Were they mated to identify the sex or for some other purpose?

Response 48: We just want to eliminate the influence of mating conditions on experimental results and noted it.

Point 49: L104 “and raised”?  I don’t know what this means.  Instead: “used”?

Response 49: Used..

Point 50: L104-105 “adaptation”?  Do you mean “acclimatization”?  How were the plants “adapted”?

Response 50: Thank you for your correction, we mean acclimization and have corrected this.

Point 51: L105 Where were they kept?

Response 51: In cages.

Point 52: L106 How was it determined that the female was mated? L103 indicates you already knew they were mated.

Response 52: We deleted this as we already knew they were mated.

Point 53: L108 What is “nutritional soil”?  Gravel or sand? Gravel is typically centimeters in diameter.

Response 53: Nutrient soil was purchased from Sichuan Luyiyuan Agricultural Technology Co., Ltd., which is mainly a mixture of humus and ordinary soil, suitable for plant growth. Changed to “nutrient soil”.

Point 54: L109 Delete “as healthy P. chinensis”.

Response 54: Done

Point 55: L109-110 I don’t understand “then placed for 6 h”.   What was placed?  Placed where?  The feeding damage was produced during 24 or 6 hours?  Was the amount of feeding assessed in some way?  The amount of feeding damage could influence the plant’s response. Delete text in parentheses.

Response 55: “Plants were placed indoors away from beetles for 6 hours” to prevent insect pheromones from remaining on the plants and affecting the experimental results and placed indoors. The feeding damage was produced during 24 hours. We have rewritten the text to explain this better.

Point 56: L112 What was your source of lighting?

Response 56: The experiment was conducted in a dark chamber.

Point 57: L115 Was the position of the treatments in the cage randomized (or at least alternated between sides of the cage for each trial)?  

Response 57: Yes, we randomized the positions to eliminate bias. We have explained this in the text.

Point 58: L116 “perching” is an odd word to use for an insect behavior (it is typically used just with birds). I can’t think of a good replacement, however.  Use “present”.

Response 58: We have eliminated the word “perching”.

Point 59: L118-120 Please provide literature citation(s),

Response 59: Done

Point 60: L120 Indicate the material of the oven bags (I think it is polyester).

Response 60: Done

Point 61: L121 How long were they dried?

Response 61: They dried for 30 minutes. Added.

Point 62: L121-122 How was the bag sealed (e.g., wrapped tightly around the stem)?

Response 61: The bag was sealed with parafilm.

Point 63: L123-124 Activated charcoal will only remove VOCs.  Merely state that the air was purified by passing through an activated charcoal filter. Replace “zero” with “purified” or “filtered”.

Response 63: Thank you for your correction, we used purified

Point 64: L125 500 µl/min is a miniscule and vastly insufficient amount of air.  Do you mean ml/min?

Response 64: Yes, I mean 500 ml/min. Thank you for your correction, we have corrected this. 

Point 65: L126-127 What do you mean by “constant volume of air”?  How was mixing obtained?

Response 65: The quantity of air is obtained by controlling the intake time. The intake time is the same, so that the volume of air is the same.

Point 66: L129 I doubt that BMILP manufactured the Tenax cartridges.  Provide the name and address of the manufacturer.

Response 66: Beijing Municipal Institute of Labor Protection, Switzerland No. 55, Taoranting Road, Xicheng District, Beijing

Point 67: L130 What was the air flow rate through the cartridges?  If you know the temperature at which the aerations were performed, indicate this. Was there a test for breakthrough through the cartridges?

Response 67: The gas flow rate of 500ml/min has been modified in the text, and no, the breakthrough through the cartridges was not tested.

Point 68: L131 Use “cartridge” not “collector”. Unless there was some specific reason for doing three sequential rinses with the same solvent, simply indicate that the cartridges were eluted with 1.5 ml hexane. Delete “solvent”.

Response 68: Done

Point 69: L133 Delete comma and “at”.

Response 69: Done

Point 70: L137 Delete “the”.

Response 70: Done

Point 71: L140 Delete “set to”.

Response 71: Done

Point 72: L143 Used for what?  I assume for making mass spectral and retention time matches to unknowns in the samples, but indicate this. Explain the quantitation method used.  Did you calculate a response curve for each compound?  Did your identifications have both a mass spectral and retention time match with your standards?

Response 72: The chemicals used to make mass spectral and retention time matches to unknowns in the samples. Identification of volatiles through NIST 08 standard library and the quantitative analysis was calculated by the peak area relative to the internal standard n-dodecane. This was added to L145-8.

Point 73: L147 How were compounds chosen for the EAG tests?

Response 73: Based on the collection and identification of volatiles experimental results, ten volatiles with higher content or that have been previously reported to attract Cerambycidae insects were selected as the compounds to be tested.

Point 74: L148 Delete “antennal potential measurement system”

Response 74: Done

Point 75: L149 Delete “the”.  Please provide a citation for the methods used.

Response 75: Done

Point 76: L150 “Paraffin” should not be capitalized.

Response 76:  Corrected

Point 77: Point L151-152 Paraffin oil is not volatile and thus did not evaporate.

Response 77: Thank you for your correction, we have noted this.

Point 78: L153 Change “a” to “the”.

Response 78: Done

Point 78: L154 Delete “the filter paper with the stimuli”.

Response 78: Done

Point 79: L155 For how long was a single antennal preparation used?

Response 79: Use the same concentration of compounds to stimulate the same antennae, and the use time should not exceed 25 minutes.

Point 80: L156 Replace “each experiment” with “trial (one antenna)” if this is correct.

Response 80: Thank you for your correction, we have corrected this.

Point 81: L158 Delete “with good activity” and change to “from active, apparently healthy B. horsfieldi”

Response 81: Done

Point 82: L160 What kind of electrodes?  An electrode fork?  If so indicate the model and manufacturer.  “Air flow rate”?  I assume this is the air flow delivering the odor?  How was the odor delivered (in glass tubing, etc.)?

Response 82: The manufacturer of the electrode fork is Syntech of Germany. The odor is passed through the Pasteur tube. This is added to the MS L167-8

Point 83: L163 Replace “EAG” with “voltage”.  Delete “with respect to the control”.

Response 83: Thank you for your correction, we have corrected this onL176.

Point 84: L165 Averaged across the entire trial?

Response 84: The average response to paraffin oil across the entire trial (L176-7)

Point 85: L167 I think you mean to say “Identifiable VOCs that increased in concentration with feeding damage were used in behavioral assays if commercially available.”

Response 85: Thank you for your correction, we have corrected this.

Point 86: L171 Were the lower surfaces inside the Y-tube lined, or was there simply a white surface beneath the Y-tube?

Response 86: The lower surface inside the Y-tube is unlined, just a white surface underneath.

Point 87: L172 Indicate source of light.

Response 87: The light source is an LED lamp, placed directly above the Y-tube

Point 88: L173 Simply “humidified”.

Response 88: Thank you for your correction, we have corrected this.

Point 89: L174 What did the odor source chambers consist of?

Response 89: The odor chamber is a 3L spherical glass instrument with interfaces at both ends (inner diameter 45mm, outer diameter 55mm)

Point 90: L177 Insert paraffin oil” after “mL”

Response 90: Thank you for your correction, we have corrected this.

Point 91: L182 What did a “choice” consist of? Presence of the insect in one of the two branches, contact with the stimulus, or something else? Were the insects prevented from touching the filter paper and how was this accomplished? How many insects of each sex were tested with each stimulus? Were insects re-used? Under what conditions were the insects kept prior to bioassays?

Response 91: Entering the side arms connecting the odor chamber and staying for more than 30 seconds be counted as a choice, and failure to enter the side arms will be counted as no choice.The test insects were not reused and maintained in insect cages under the same conditions. Added to L196-8

Point 92: L184 Change to “Dilutions of standard compounds (2 mL at 10 mg/ml) were evenly coated...”. How were the compounds selected for testing?

Response 92: We selected compounds that elicit behavioral responses. Thank you for your correction, we have corrected this.

Point 93: L185 How were the branches presented?  Were they simply lying horizontally on the floor or were they supported in some way?  Did the branches include foliage?

Response 93: Branches without foliage placed vertically against the six corners of the hexagonal insect cage (1 m height, 1 m per side) L204-6.

Point 94: L187 Write: “…hexaganol insect cage (1 m height, _ m per side)…” and delete next sentence. Indicate what was your control in the first sentence of the paragraph.

Response 94: The control is untreated branches without foliage. Thank you for your correction, we have corrected this.

Point 95: L188 Replace “insect” with “the”.  I don’t understand “the feeding notch on the branch was recorded”. It might be sufficient to indicate that the surface area of feeding damage was recorded.

Response 95: We measured and recorded the feeding area using parchment paper and grid paper (1 mm2 per grid). Thank you for your correction, we have amended this.

Point 96: L190 How was the area measured on the graph paper?

Response 96: There are small grids of 1 mm2 on the graph paper. We covered the surface with parchment paper and traced the pattern to calculate the area. L209-210.

Point 97: L193-194 What were these tests used to compare (what was the dependent variable)?  The LSD test presumably was used with an ANOVA; please explain this.  Please indicate whether a correction was made to the LSD test for experiment-wise error (i.e., a Bonferroni correction).  Normally, an LSD test is not considered acceptable without such a correction. Same for L195-196.  How was a chi-square test – used for count data – used in analysis of the EAG data?

Response 97: We referred to the analytical methods of other researchers. Chi-square test was used to compare between males and females, and one-way ANOVA were used to compare the response values of each compound.

Point 98: L193-196 How were tests done for the Y-tube assays and the quantities of plant volatiles?

Response 98: T test was conducted to evaluate the Y-tube assays.

Point 99: L202 Use “host plant choice tests” instead of “perching selectivity experiments”

Response 99: Done

Point 100: L204 If a t-test was performed, then it requires a t-distribution, not an F-distribution.

Response 100: Thank you for your correction, we have corrected this.

Point 101: L208 Change “perching selectivity” to “host preference tests”.  Change “to” to “with”

Response 101: Done

Point 102: L209-210 Change “P. chinensis” to “one or the other plant treatment”.

Response 102: Done

Point 103: L212 Delete “number”.

Response 103: Done.

Point 104: L213-217 This information should go to the methods section.  If the mass spectra of all compounds were matched to spectra of standards, you don’t need to mention use of the NIST library.  Compounds identified with the NIST library alone should be indicated as tentative.

Response 104: Thank you for your correction. We have moved this info to L145-8 in methods.

Point 105: L214 Delete “, the components…computer,”

Response 105: Done.

Point 106: L216-217 Delete “detected…classes:”

Response 106: Done.

Point 107: L217 Benzene is a specific compound; the class is “aromatics”.  These compounds (most of those listed as “benzene”) are common contaminants of hexane.  If these (and other compounds) were in your extraction solvent (which is very likely) they should be removed from the table.  An extract of an unused cartridge should have been used as a control to eliminate these compounds from the data.  I strongly suspect that toluene, all three xylenes, ethylbenzene, and styrene, as well as several other compounds in the list, are solvent contaminants.

Response 107: The data was checked and ten solvent contaminants were removed. Thank you for your correction, we have corrected this.

Point 108: L218 Make reference to table 2.

Response 108: Done.

Point 109: L218-219  Condense these two sentences to “β-ocimene was the predominant compound in healthy plants and myrcene in damaged plants.”  

Response 109: Done

Point 110: Table 2:  Units of “Content” need clarification.  Per liter extraction solvent or air drawn into the cartridges?  It is more typical for this kind of data to be reported as amount per sample.

Response 110: Thank you for your correction, we have corrected this as per L extraction solvent

Point 111: L221 Delete “identical”.

Response 111: Done.

Point 112: L227 Change “which have not been” to “that were not”.

Response 112: Done.

Point 113: L231 Delete comma and “the”.

Response 113: Done.

Point 114: L232 Change “highest” to “present in highest concentrations”.

Response 114: Done.

Point 115: L233 Many compounds listed as terpenes are not. Terpenes are unsaturated compounds with carbons in multiples of five (5,10,15 etc.)

Response 115: Thank you for your correction, we have corrected this.

Point 116: L234-235 An LSD test is used for multiple comparisons. Only two values are being compared here, it seems. A t-test is appropriate. With just a single contrast, indicate a difference with an asterisk rather than significance lettering. Change “in different states” to “of the two treatments”.

Response 116: Thank you for your correction, we have corrected this.

Point 116: L242 What is meant by “measurable”?

Response 116: Detectable. Thank you for your correction, we have corrected this.

Point 117: L257-260 If differences were not statistically significant, they should not be mentioned. Also, if there were no differences between males and females, male and female data should be pooled and data presented and discussed for sexes combined.

Response 117: Thank you for your suggestions, we have corrected this.

Point 118: L264 “differences”

Response 118: Done

Point 119: L265 Insert “, respectively,” after “female”

Response 119: Done

Point 120: L265-267 Delete this sentence since (according to the figure) there were no differences between males and females for any single compound.

Response 120: Thank you for your suggestion, we have corrected this.

Point 121: Figure 2. There appear to be no differences between males and females.  If this is true, remove the “ns” markers.  It would also be helpful to pool data from males and females if sexes did not differ.  The graph would be easier to interpret, and you might detect more significant differences among compounds.

Response 121: Thank you for your suggestions, we have done this.

Point 122: L270 Change “which…value of” to “which elicited the strongest EAG responses from”

Response 122: Done

Point 123: L271-272 Change this sentence to “Selection of the treatment over the control olfactometer arm by females was greatest for (Z)-3-hexen-1-ol (80% ± 3% SE), β-ocimene (83% ± 2% SE), …” etc. for this and other such sentences in the paragraph. The percentages should have only two significant figures.

Response 123: Done

Point 124: L274-277 It does not seem the stats were appropriate here.  A t-test was not appropriate if percentage responses to the two arms were compared, since these values are not independent (the value of one is dependent on the value of the other).  Also, the data are counts of responses, and a t-test is not appropriate for these kinds of data. The correct test for 2-choice olfactometers used with single insects is a G-test or exact probabilities test where the proportions of insects responding to each arm (e.g., 2 vs. 8) is contrasted with the null hypothesis of no preference (5:5)

Response 124: We use t-test to analyze the behavioral response of B. horsfieldi to compounds data according to the literature.

Point 125: L275 Change “selected for” to “selecting”

Response 125: Done

Point 126: L276 Change “were” to “was”.

Response 126: Done

Point 127: L285 Change “Olfactory behavior of” to “Response in a Y-tube olfactometer by”

Response 127: Done

Point 128: Figure 3. Delete “<---> paraffin oil” and “odor source combinations” from the left-hand titles. Instead, write “test compound” above the left-hand bars and “paraffin oil” above the right-hand bars at the top of each plot

Response 128: Figure was revised.

Point 129: L290-292 This information should only be given in the methods section.   An explanation of why these specific compounds were chosen should be provided.

Response 129: We selected these specific compounds that could induce behavioral responses. This content has been supplemented in the methods section. Thank you for your correction, we have corrected this.

Point 130: L292 Change “feeding area” to “amount of bark removed by feeding”

Response 130: Done

Point 131: L293 Delete “, which…mm2,” Data are sufficiently presented in the graph and don’t need repetition here.  Replace first comma with “and”.

Response 131: Done

Point 132: L295-298 See comment for line 293.

Response 132: Done

Point 133: L300 “lowercase” to “lower case”

Response 133: Done

Point 134: L301 Change “significantly” to “significant”.  Change “feeding intensity of” to “bark area consumed by”.  Change “to” to “on”.

Response 134: Done.

Point 135: Figure 4:  On Y-axis label, change “Feeding area” to “Area of bark consumed”

Response 135: Done.

Point 136: L304 Delete “to perch on”.

Response 136: Done

Point 137: L305 Provide a citation of these previous observations, if published.

Response 137: Done

Point 138: L306 Replace “the selection…plants” with “female and male B. horsfieldi selected damaged P. chinensis 3-fold more often than undamaged plants.”

Response 138: Thank you for your correction, we have corrected this.

Point 139: L304-371 The discussion section is mostly repetition of results and repetition of information already presented sufficiently in the introduction.  The data is very interesting and deserves sufficient commentary so the reader can appreciate this. You leave many important topics unaddressed. For example, do your results change our understanding of HIPVs?  Have similar results been found for other cerambycids? How are they similar and how different?  Are all or some of the findings novel and in what way? Why might this be?  Have these same compounds been found to influence beetle behavior in other cerambycid-host systems? What was concluded about their ecological role in these systems?  What is known about the compounds that produced responses (are they common or rare in plants, and how might this impact behaviors in nature)? Why would some be repellant and others attractive (e.g., why would limonene be special)– both generally and in the specific case of B. horsfieldi?  How could your findings be used in applications?  Are there specific problems or needs in pest management that could be addressed for B. horsfieldi with your findings?  Should HIPVs such as you found mediate beetle aggregation on damaged hosts and influence spatial distribution of damage? You don’t need to answer all of these questions, but addressing those you feel most important will greatly improve the value of your work to readers. Your hard and careful work in the laboratory deserves such explanations.  Only repeat the reporting of results when necessary to support your answers to the questions above.

Response 138: Thanks for your suggestion. Our result found that the HIPVs of healthy P. chinensis were different from those of feeding-damaged tree, and the contents of the same volatiles were different. The adults of B. horsfieldi prefer to feed on damaged P. chinensis for these differences in plant volatiles. 

Similar results have been found in cerambycids, such as Monochamus alternatus and Arhopalus tristis. Ovipositing female M. alternatus prefer stressed Pinus massoniana over healthy trees, for that α-pinene, β-pinene and D-limonene were more abundant in stressed trees than in healthy ones which has a good attraction to the M. alternatus [3]. Female A. tristis had a strong preference for the volatiles of burnt vs. unburnt pine bark, for those monoterpenes have been isolated from the volatiles of smouldering wood and monoterpenes attracted female A. tristis. [4].

We only detected (Z)-3-Hexen-1-ol in feeding-damaged trees of P. chinensis, but not on healthy trees. B. horsfieldi showed the most significant EAG response to (Z)-3-Hexen-1-ol. At the same time, the content of β-ocimene in damaged plants increased significantly. (Z)-3-Hexen-1-ol and β-ocimene significantly promoted the feeding of B. horsfieldi. The content of D-Limonene in damaged plants was significantly lower than that in healthy plants. This may be the main reason why B. horsfieldi prefer damaged P. chinensis, which has not been reported before. The specific mechanism of action needs to be further studied. Our research is helpful to provide reference for the research and development of sex attractants for B. horsfieldi.

In EAG assays, the antennae of B. horsfieldi adults responded strongly to (Z)-3-hexen-1-ol, β-ocimene, 3-carene, γ-terpinene, D-limonene, myrcene, and α-phellandrene. The seven compounds with strong EAG response are common in nature.  (Z)-3-hexen-1-ol has a good attraction for Empoasca onukii and some cerambycids [5,6]. β-ocimene was reported to attract female Aphidius ervi. And β-ocimene has also been shown to increase both mating and oviposition rates in Hyphantria cunea [7]. α-pinene, 3-carene, and D-limonene were identified as major constituents in the essential oil of roots of Angelica archangelica, which showed antifungal activity against fungi. Moreover, the bicyclic monoterpene 3-carene is reported to be an attractant for Dendroctonus and Hylurgops [8]. γ-terpinene was found to have trypanocidal effect [9] and lure effect. α-phellandrene would cause cuticular damage of larvae of Lucilia cuprina [10,11]. D-limonene has been reported to have antibacterial and antiviral activity [12,13]. Myrcene is a attractant of many other bark beetle species, exhibited a strong repellent (or inhibitory) effect on I. typographus and Aromia bungii females [14,15]. In summary, these compounds play important roles in nature and influence the behavior of B. horsfieldi.

The compounds we identified are repellent or attractant to B. horsfieldi, which are related to the odor-binding proteins and olfactory receptors, and need further study. The effect of HIPV on B. horfieldi in forest remains uncertain. Different compounds mixed in certain proportions may enhance the attraction to B. horsfieldi. Our study provides a theoretical framework for further research on the effective attractant components of B. horsfieldi and for the development of attractants or repellents. The HIPV we found can mediate the beetle aggregation on the damage hosts, thus influence the spatial distribution of damage. Altogether, our study is meaningful.

The discussion section has been completely re-written to include a lively discussion for the readers as we answered many of the suggested questions that would make our careful labwork more interesting and revelent to readers. 

Thank you to the reviewers for their revised comments, which will be of great benefit to our future paper writing and test specifications. I wish you all the best!

Dr. Jianting Fan

Associate Professor

School of Forestry and Bio-technology

Zhejiang A & F University

Wusu Street #666

Lin’an, Hangzhou, Zhejiang, 311300

CHINA

Email: fanjt@zafu.edu.cn

  1. Fraga D.F.; Parker J.; Busoli A.C.; Hamilton G.C. Behavioral responses of predaceous minute pirate bugs to tridecane, a volatile emitted by the brown marmorated stink bug. J. Pest Sci. 2017, 90, 1107–1118. https://doi.org/10.1007/s10340-016-0825-9
  2. Anastasaki E.; Drizou F.; Milonas P.G. Electrophysiological and Oviposition Responses of Tuta absolutaFemales to Herbivore-Induced Volatiles in Tomato Plants. Chem. Ecol. 2018, 44, 288–298. https://doi.org/10.1007/s10886-018-0929-1
  3. Fan J.T.; Sun J.H. & Shi J. Attraction of the Japanese pine sawyer, Monochamus alternatus, to volatiles from stressed host in China. For. Sci.2007, 64, 67–71. https://doi.org/10.1051/forest:2006089
  4. Zhong J.P.; Xiao X.Y.; Jin M.X.; Tu Y.G.; Xie G.A.; Wang W.H. A Bibliometric Analysis of Researches of Batocera horsfieldi. Disaster Sci. 2022, 45, 194-198.
  5. Bian L.; Cai X.M.; Luo Z.X.; Li Z.Q.; Xin Z.J.; Chen Z.M. Design of an Attractant for Empoasca onukii(Hemiptera: Cicadellidae) Based on the Volatile Components of Fresh Tea Leaves. Econ. Entomol. 2018, 2, 629-636. doi: 10.1093/jee/tox370. PMID: 29361007.
  6. Nehme M.E.; Keena M.A.; Zhang A.; Baker T.C.; Hoover K. Attraction of Anoplophora glabripennisto Male-Produced Pheromone and Plant Volatiles. Entomol. 2009, 38, 1745–1755. https://doi.org/10.1603/022.038.0628
  7. Tang R.; Zhang F.; Zhang Z.N. Electrophysiological Responses and Reproductive Behavior of Fall Webworm Moths (Hyphantria cuneaDrury) are Influenced by Volatile Compounds from Its Mulberry Host (Morus alba). Insects. 2016, 7, 19. doi: 10.3390/insects7020019. PMID: 27153095; PMCID: PMC4931431.
  8. Kelsey R.G.; Westlind D.J. Attraction of red turpentine beetle and other Scolytinae to ethanol, 3-carene or ethanol + 3-carene in an Oregon pine forest. For. Entomol.2018, 20.
  9. Baldissera M.D.; Grando T.H.; Souza C.F.; Gressler L.T.; Stefani L.M.; da Silva A.S.;  Monteiro S.G. In vitro and in vivo action of terpinen-4-ol, γ-terpinene, and α-terpinene against Trypanosoma evansi. Experimental parasitology2016, 162, 43–48. https://doi.org/10.1016/j.exppara.2016.01.0041
  10. Chaaban, A., Richardi, V. S., Carrer, A. R., Brum, J. S., Cipriano, R. R., Martins, C. E. N., Navarro-Silva, M. A., Deschamps, C., & Molento, M. B. (2018). Cuticular damage of Lucilia cuprina larvae exposed to Curcuma longa leaves essential oil and its major compound α-phellandrene. Data in brief, 21, 1776–1778. https://doi.org/10.1016/j.dib.2018.11.001
  11. Chaaban A.; Richardi V.S.; Carrer A.R.; Brum J.S.; Cipriano R.R.; Martins C.E.N.; Navarro-Silva M.A.; Deschamps C.; Molento M.B. Cuticular damage of Lucilia cuprinalarvae exposed to Curcuma longa leaves essential oil and its major compound α-phellandrene.  2018, 21, 1776–1778. https://doi.org/10.1016/j.dib.2018.11.001
  12. Andriotis E.; Papi R.M.; Paraskevopoulou A.; Achilias D.S. Synthesis of D-Limonene Loaded Polymeric Nanoparticles with Enhanced Antimicrobial Properties for Potential Application in Food Packaging. Nanomaterials (Basel). 2021, 11, 191. doi: 10.3390/nano11010191. 
  13. Corrêa A.N.R.; Weimer P.; Rossi R.C.; Hoffmann J.F.; Koester L.S.; Suyenaga E.S.; Ferreira C.D. Lime and orange essential oils and d-limonene as a potential COVID-19 inhibitor: Computational,in chemico, and cytotoxicity analysis. Food bioscience 2023, 51, 102348. https://doi.org/10.1016/j.fbio.2022.102348
  14. Bozsik G.; Molnár B.P.; Domingue M.J.; SzÅ‘cs G. Changes to volatile profiles of arborvitae, Thuja occidentalis, from drought and insect infestation: olfactory cues for the cypress bark beetle, Phloeosinus aubei. Chemoecology2023, 33, 113–124. https://doi.org/10.1007/s00049-023-00389-9
  15. Cao D.; Liu J.; Zhao Z.; Yan X.; Wang W.; Wei J. Chemical Compounds Emitted from Mentha spicata Repel Aromia bungii Insects2022, 13, 244. https://doi.org/10.3390/insects13030244

Reviewer 2 Report

Comments and Suggestions for Authors

Plant volatiles released from plants are able to attract or repel insects, which possibly can be considered as a strategy for pest management.  Batocera horsfieldi Hope feeds many different species of plants, causing severe damage in many aspects. Pest control of this insect will definitely benefit much for the environment as well as economy directly and indirectly.  This study tested how Batocera horsfieldi preferred to different plants using feeding-damaged or healthy plant Pistacia chinensis Bunge. Authors also isolated and identified many different compounds from these two different states of plants. As results of the following assays, adults of Batocera horsfieldi showed different responses to those identified compounds using electro antennography and also presented distinct behavioral selection to these compounds with sexual dimorphism between male and mated females. This study has potential guide on how to manage pest in an environment-friendly way with possible high efficacy. Overall, experiments were conducted well. However, this manuscript can be further improved to make it more understandable for readership. The English improvement is required.

1. The abstract can be improved to make it clear and concise. For example, lines 21-23, this sentence can be improved to be more understandable, and its current expression is confusing. 

2. line 25, “on” should be replaced by “to”, it will be better.

3.line 27, the expression “feeding preference tests were carried out” did not have a good connection with the text. 

4.line 36, I am confused on what is “attractant activity”. Do authors have other phrase?

5. line 39, how to understand “promote the intensity of feeding by B. horsfieldi” ? Correct to be clear.

6. line 40, add “that” before D-limonene.

7. line 45, what is “an important wood-boring pest”? Is it good to use “important”?

8. line 59, the expression “allow them to be protected themselves again “ is confusing. Change it to be more understandable. 

9. line 60, “is” should be “are”.

10. line 62, “responds” should be “respond”.

11. line 86-87, the expression is not understandable. 

12. line 216, delete “belonging to”.

13. line 218, “β-ocimene was the highest”, delete “the”, same situation for line 220, authors should read through the manuscript to revise these grammas, many can be found in the text, such as other “the”  in line 228. 

14.  line 227, add comma before “which”, current expression is very different.

15. lines 239-241, the sentences are not understandable and confusing. 

16. line 242, “test” should be “tested”.

17. line 245, the content in the bracket is not consistent with that in lines 250 and 255.

18. line 259, “but no” should be “but there is no”. Go through the manuscript to change others like this, many can be found.

19. line 275, “ selected for” should be “selecting”. Same problems for line 279. Go through the manuscript to find others.

20. line 287, add indicates between “ns” and “no”.

21. line 319 to 321, the expression can be improved to be better.

22. line 334 to 335, EAD response is for the insects, it is not for the compound. Change the sentence to make it clear.

23. line 338, difficult to understand “but was the weakest”.

24. line 348-349, “and adults of both sexes had 348 obvious repellant effects to D-limonene” is confusing.

25. line 26, “attractive single compound” to “single attractive compound”.

26. line 365, add “that” between “and” and “D”.

27. line 367, revise “If”.

28. line 370, change “provide” to “provides”.

Comments on the Quality of English Language

English can be improved to make the manuscript better.

Author Response

Thanks to the experts for their careful review, we have made careful modifications according to modification opinions proposed by the experts.

The revised comments are as follows:

Point 1: The abstract can be improved to make it clear and concise. For example, lines 21-23, this sentence can be improved to be more understandable, and its current expression is confusing. 

Response 1: Done

Point 2: line 25, “on” should be replaced by “to”, it will be better.

Response 2: Done

Point 3: line 27, the expression “feeding preference tests were carried out” did not have a good connection with the text. 

Response 3: Thank you for your suggestion, we have corrected this. It now reads “Host plant choice tests show that B. horsfieldi prefers to choose feeding-damaged P. chinensis rather than healthy trees.”

Point 4: line 36, I am confused on what is “attractant activity”. Do authors have other phrase?

Response 4: Thank you for your suggestion, we have corrected this. Sentence now reads "Y-tube behavioral experiments showed that four compounds attracted mated females ((Z)-3-hexen-1-ol, β-ocimene, 3-carene, and α-phellandrene), two compounds ((Z)-3-hexen-1-ol and α-phellandrene) attracted males, and adults of both sexes avoided D-limonene.”

Point 5: line 39, how to understand “promote the intensity of feeding by B. horsfieldi” ? Correct to be clear.

Response 5: Thank you for your suugestion, we have corrected this. The sentence now reads “Feeding bioassays showed that (Z)-3-hexen-1-ol and β-ocimene could promote feeding of B. horsfieldi and that D-limonene inhibited this response.”

Point 6: line 40, add “that” before D-limonene.

Response 6: Done.

Point 7: line 45, what is “an important wood-boring pest”? Is it good to use “important”?

Response 7: Thank you for your suggestion to change this, we have re-written  this as “Batocera horsfieldi Hope (Coleoptera: Cerambycidae) is one of the major native forestry trunk borers in China...”

Point 8: line 59, the expression “allow them to be protected themselves again “ is confusing. Change it to be more understandable. 

Response 8: We deleted the sentence.

Point 9: line 60, “is” should be “are”.

Response 9: Done

Point 10: line 62, “responds” should be “respond”.

Response 10: Changed.

Point 11: line 86-87, the expression is not understandable. 

Response 11: We have deleted the sentence from the manuscript.

Point 12: line 216, delete “belonging to”.

Response 12: Done

Point 13: line 218, “β-ocimene was the highest”, delete “the”, same situation for line 220, authors should read through the manuscript to revise these grammas, many can be found in the text, such as other “the”  in line 228. 

Response 13: We have carefully copyedited the manuscript to correct these errors.

Point 14:  line 227, add comma before “which”, current expression is very different.

Response 14: Done

Point 15: lines 239-241, the sentences are not understandable and confusing. 

Response 15: Thank you for your correction, we have  revised this section.

Point 16: line 242, “test” should be “tested”.

Response 16: Done

Point 17: line 245, the content in the bracket is not consistent with that in lines 250 and 255.

Response 17: Thank you for your suggestion, we have revised this section.

Point 18: line 259, “but no” should be “but there is no”. Go through the manuscript to change others like this, many can be found.

Response 18: Thank you for your correction, we have carefully copyedited the manuscript to correct these errors.

Point 19: line 275, “ selected for” should be “selecting”. Same problems for line 279. Go through the manuscript to find others.

Response 19: Thank you for your suggestion, we have corrected this throuout the manuscript.

Point 20: line 287, add indicates between “ns” and “no”.

Response 20: Thank you for your suggestion, we have changes this.

Point 21: line 319 to 321, the expression can be improved to be better.

Response 21: Thank you for your correction, we have revised this section.

Point 22: line 334 to 335, EAD response is for the insects, it is not for the compound. Change the sentence to make it clear.

Response 22: Thank you for your suggestion, we have corrected this to correctly indicate the insect response to the compound..

Point 23: line 338, difficult to understand “but was the weakest”.

Response 23: Thank you for your correction, we have corrected this.

Point 24: line 348-349, “and adults of both sexes had 348 obvious repellant effects to D-limonene” is confusing.

Response 24: This sentence was deleted.

Point 25: line 26, “attractive single compound” to “single attractive compound”.

Response 25: Done

Point 26: line 365, add “that” between “and” and “D”.

Response 26: Done

Point 27: line 367, revise “If”.

Response 27: Done

Point 28: line 370, change “provide” to “provides”.

Response 28: Done

Thank you to the reviewers for their revised comments, which will be of great benefit to our future paper writing and test specifications. I wish you all the best!

Dr. Jianting Fan

Associate Professor

School of Forestry and Bio-technology

Zhejiang A & F University

Wusu Street #666

Lin’an, Hangzhou, Zhejiang, 311300

CHINA

Email: fanjt@zafu.edu.cn

Round 2

Reviewer 1 Report

Comments and Suggestions for Authors

L14  I suggest inserting “in China” after “forestry pest”

L21 “Some phytophagous insects…”  not all

L32 “significant differences in quantities of 5 terpenes”

L63  Delete “the”

L72 Delete “host”.  Change “localization”  to “host finding and selection”

L97 “feeding” can be deleted

L99-101  As written now, it sounds as if only mated females were used in subsequent experiments.  When mating was observed, were both sexes moved to new cages?  Were these mixed-sex cages?

L110 Delete “selective”

L115  Change “choosing”  to “present on”

L124  Change to  “(air without volatile organic compounds (VOCs))”

L125 Delete “oven bag”

L126 Delete second “the”

L128 Change “over” to “through”  Change “absorbent” to “adsorbent”

L131 REpolace “pure chromatographic” to “chromatography grade”

L143 Replace “amu”  with “m/z”

Table 1 Dodecane appears to be repeated.  For the EAD tests it would be useful to know the enantiomer of the chiral compounds.  For example:  (-)-alpha-pinene, (-)-beta-pinene.  Olfactory sensitivities can be different for different enantiomers.   If it is not written on the bottle, don’t worry about it.

L147-148  Since the data are reported  as µg/L, you should have calculated response factors to equate ion  abundance with mass for each individual compound.  Indicate whether response factors were calculated how. 

L152 Change to “Based on results of volatiles collections, “

L153 Change “higher content” to “greatest abundance”

L154 delete “insects”

L155 Delete “of different concentrations”

L156 Delete “the”

L157-158  Delete “non-volatile”

L159  Change “a” to “the”

L160 Insert “alone” after “oil”  Insert “each” before “trial”

L162-163 Delete delete “from healthy B. horsefieldi adults “

L164-165 Change to “Every antenna was exposed to all compounds and concentrations, with antennal preparations not used for more than 25 min.”  if this is correct.

L165   Change to “The tip…fork”  to “The two cut ends of the antenna were immersed in electroconductivity gel on opposite conductors of an electrode fork…”

L167 -169  This information (except for the final clause) was given earlier, so delete this text.  What I was looking for was something like “the tip of the pipette was inserted through a hole in a piece of glass tubing that delivered a continuous flow of purified, humidified air at __ml/min over the antennal preparation”.  I do not know what a “console” is. 

L170 174  Delete “After the baseline signal stabilized and measurements began”  and rewrite sentence simply as “The pulse stimulation air flow was 120 mL/min, and the duration of stimulation was 0.5 sec.”  Delete next sentence.

L175 Insert “to test compounds” after “responses”.  Delete “the control, which is”

L180-182  Delete the “interface diameters” from the specifications

L186  Delete “shape”

L190 insert “in paraffin oil” after “pinene”

L175  Was the sex of the beetles known?

L202 Change to “Dilutions (2 mL at 10 mg/mL paraffin oil) of selected compounds that elicited behavioral responses were evenly coated..”   Was paraffin oil the solvent? 

L203 change “without”  to “lacking”

L204  Insert period after  “diameter)” Delete “and the” and begin next sentence with “Treated and control (untreated) branches…”   The control branches were not treated with the solvent for the test compound?

L206  Delete “, and the bottom surface…length”

L206-207  Unless the grid had a purpose in the experiment, delete this sentence.

Table 2.  I cannot find any information on  2,3-Dimethyl-1,3-butadiene being a plant compound.  Please check again that this is not a solvent contaminant.  

Statistical analyses – problems still need to be4 addressed:

1)       An independent sample t-test is not appropriate for choice tests (the Y-tube experiments or the plant choice tests).  These experiments require an exact probabilities test or a G-test. 

2)      Multiple comparisons tests with an LSD are not acceptable without a correction for experiment-wise error such as a Bonferroni test.

3)      Statistics for the quantitative volatiles analyses are not explained

L246 Change “had not been” to “was”

L251 Change “had no significant difference” to “did not differ among compounds”

L252-254  If these are “relative response values”  then  a value of 1 is that of the paraffin oil blank;  hence it is not possible that limonene elicited a significant response if its relative response ratio was less than 1.   A significant response is one that has a statistically greater value than 1 – correct?

L257-258  Male responses  to Z-3-hexen-1-ol were not higher than beta-ocimene.

L267  Were there significant differences or not?  L267-270 indicates there were differences and L270-271 states that there weren’t any.

L275 LSD test alone is not acceptable; P values must receive a Bonferroni correction or equivalent.

L280 Change “on” to “with”

L280-283 Delete sentence.  It largely duplicates the next sentence and the information on the percentages is evident in the figure.

L280-283 Delete sentence.  It largely duplicates the next sentence and the information on the percentages is evident in the figure.

L285 “were  all significantly higher than that of the control group”  to  “was significantly higher than those selecting controls”

L286-287 Delete sentence. Same explanation as above.   Change “selected” to “selecting”.

L300 Delete comma.

L302 Change “of” to “on”

L310 Delete “to choose”

L312 Change “selectivity” to “choice”

L322 Delete “the”

L323 Change “periods” to “plants”

L328  Change “highest” to “most abundant”

L334 Replace “for that” with “and”

L343 Change “is hig” to “was particularly strong”

L349-351 Delete sentence

L366 Insert “may” before “play”

L371-378 Delete sentences.

L379-381  Delete sentence.

L388 Change “provides” to “should provide”

Comments on the Quality of English Language

English was substantially improved, but issues still existed that I tried to fix. 

Author Response

Dear Editor:

Thanks to the experts for their careful review, we have made careful modifications according to modification opinions proposed by the experts.

The revised comments are as follows:

Point 1:L14  I suggest inserting “in China” after “forestry pest”

Response 1: Thank you for your correction, we have corrected this.

Point 2: L21 “Some phytophagous insects…”  not all

Response 2:  Done

Point 3: L32 “significant differences in quantities of 5 terpenes”

Response 3:  Done

Point 4: L63  Delete “the”

Response 4:  Done

Point 5: L72 Delete “host”.  Change “localization”  to “host finding and selection”

Response 5: Thank you for your correction, we have corrected this.

Point 6: L97 “feeding” can be deleted

Response 6: Done

Point 7: L99-101 As written now, it sounds as if only mated females were used in subsequent experiments.  When mating was observed, were both sexes moved to new cages?  Were these mixed-sex cages?

Response 7: Yes, both sexes were moved to new cages and raised separately. We have corrected this.

Point 8: L110 Delete “selective”

Response 8: Thank you for your correction, we have corrected this.

Point 9: L115  Change “choosing”  to “present on”

Response 9: Done

Point 10: L124 Change to “(air without volatile organic compounds (VOCs))”

Response 10: Thank you for your correction, we have corrected this.

Point 11: L125 Delete “oven bag”

Response 11: Done

Point 12:L126 Delete second “the”

Response 12: Done.

Point 13: L128 Change “over” to “through” Change “absorbent” to “adsorbent”

Response 13: Done.

Point 14: L131 REpolace “pure chromatographic” to “chromatography grade”

Response 14: Done.

Point 15: L143 Replace “amu” with “m/z”

Response 15: Thank you for your correction, we have corrected this.

Point 16:Table 1 Dodecane appears to be repeated.  For the EAD tests it would be useful to know the enantiomer of the chiral compounds.  For example:  (-)-alpha-pinene, (-)-beta-pinene.  Olfactory sensitivities can be different for different enantiomers.   If it is not written on the bottle, don’t worry about it.

Response 16:  Thank you for your correction, we have corrected this. The enantiomers of chiral compounds are not written on the bottle.

Point 17: L147-148  Since the data are reported  as µg/L, you should have calculated response factors to equate ion  abundance with mass for each individual compound.  Indicate whether response factors were calculated how. 

Response 17: Thank you for your correction, we have done it. We used n-dodecane as a qualitative internal standard. And compounds of each sample were eluted from the Supelco pack with 500 µL hexane containing 50 ng/µL n-dodecane. Quantitative analysis of volatiles were calculated from peak areas based on the response of the internal standard n-dodecane and the response factors for standard compounds.

Point 18: L152 Change to “Based on results of volatiles collections, “

Response 18: Done.

Point 19: L153 Change “higher content” to “greatest abundance”

Response 19: Thank you for your correction, we have corrected this.

Point 20: L154 delete “insects”

Response 20: Done.

Point 21:L155 Delete “of different concentrations”

Response 21: Done.

Point 22: L156 Delete “the”

Response 22: Done.

Point 23:L157-158  Delete “non-volatile”

Response 23: Done.

Point 24: L159  Change “a” to “the”

Response 24: Done.

Point 25:L160 Insert “alone” after “oil”  Insert “each” before “trial”

Response 25: Thank you for your correction, we have corrected this.

Point 26: L162-163 Delete delete “from healthy B. horsefieldi adults “

Response 26: Done.

Point 27: L164-165 Change to “Every antenna was exposed to all compounds and concentrations, with antennal preparations not used for more than 25 min.”  if this is correct.

Response 27: Thans for your suggestion. We have revise this sentence to  Every antenna was exposed to same concentration of different compounds, with not used for more than 25 min.”

Point 28: L165   Change to “The tip…fork”  to “The two cut ends of the antenna were immersed in electroconductivity gel on opposite conductors of an electrode fork…”

Response 28: Thank you for your correction, we have corrected this.

Point 29: L167 -169 This information (except for the final clause) was given earlier, so delete this text.  What I was looking for was something like “the tip of the pipette was inserted through a hole in a piece of glass tubing that delivered a continuous flow of purified, humidified air at __ml/min over the antennal preparation”.  I do not know what a “console” is. 

Response 29:  Thank you for your correction, we have corrected this. The tip of glass Pasteur pipette tube was inserted through a hole in a piece of glass tubing that delivered a continuous flow of purified, humidified air at 600 ml/min over the antennal preparation.

Point 30: L170 174  Delete “After the baseline signal stabilized and measurements began”  and rewrite sentence simply as “The pulse stimulation air flow was 120 mL/min, and the duration of stimulation was 0.5 sec.”  Delete next sentence.

Response 30: Done.

Point 31: L175 Insert “to test compounds” after “responses”.  Delete “the control, which is”

Response 31: Thank you for your correction, we have corrected this.

Point 32: L180-182  Delete the “interface diameters” from the specifications

Response 32: Thank you for your correction, we have corrected this.

Point 33:L186  Delete “shape”

Response 33: Thank you for your correction, we have corrected this.

Point 34: L190 insert “in paraffin oil” after “pinene”

Response 34: Done.

Point 35: L175 Was the sex of the beetles known?

Response 35: Yes, all beetle sexes were known

Point 36: L202 Change to “Dilutions (2 mL at 10 mg/mL paraffin oil) of selected compounds that elicited behavioral responses were evenly coated..”   Was paraffin oil the solvent? 

Response 36: Thank you for your correction, we have corrected this. Yes, paraffin oil was a solvent.

Point 37: L203 change “without”  to “lacking”

Response 37: Thank you for your correction, we have corrected this.

Point 38: L204 Insert period after “diameter)” Delete “and the” and begin next sentence with “Treated and control (untreated) branches…”   The control branches were not treated with the solvent for the test compound?

Response 38: Thank you for your correction, we have corrected this. Yes, the control branches were not treated with the solvent of the test compound.

Point 39: L206  Delete “, and the bottom surface…length”

Response 39: Done.

Point 40: L206-207  Unless the grid had a purpose in the experiment, delete this sentence.

Response 40: Done.

Point 41: Table 2.  I cannot find any information on  2,3-Dimethyl-1,3-butadiene being a plant compound.  Please check again that this is not a solvent contaminant

Response 41: Thank you for your correction, we have corrected this. 2,3-Dimethyl-1,3-butadiene may be solvent contaminant

Statistical analyses – problems still need to be4 addressed:

Point 42: An independent sample t-test is not appropriate for choice tests (the Y-tube experiments or the plant choice tests).  These experiments require an exact probabilities test or a G-test. 

Response 42: Thanks for your suggestion, we have revised this. We used chi-square test to compare groups for choice tests again.

Point 43:  Multiple comparisons tests with an LSD are not acceptable without a correction for experiment-wise error such as a Bonferroni test.

Response 43: Thanks for your suggestion, we have modified this. Multiple comparisons tests were performed with Tukey’s test and Bonferroni correction.

Point 44: Statistics for the quantitative volatiles analyses are not explained

Response 44: Thanks for your suggestion, We have explained statistics for the quantitative volatiles analyses in results(3.2. Collection and identification of volatile compounds).

Point 45: L246 Change “had not been” to “was”

Response 45: Done

Point 46: L251 Change “had no significant difference” to “did not differ among compounds”

Response 46: Done

Point 47: L252-254  If these are “relative response values”  then  a value of 1 is that of the paraffin oil blank;  hence it is not possible that limonene elicited a significant response if its relative response ratio was less than 1.   A significant response is one that has a statistically greater value than 1 – correct?

Response 47: Yes, it is not possible that limonene elicited a significant response when its relative response ratio was less than 1.

Point 48: L257-258  Male responses  to Z-3-hexen-1-ol were not higher than beta-ocimene.

Response 48: At the 10 mg/mL concentration, males of B. horsfieldi responses  to Z-3-hexen-1-ol were higher that of the other compounds on males.

Point 49: L267  Were there significant differences or not?  L267-270 indicates there were differences and L270-271 states that there weren’t any.

Response 49:  There were no significant differences between male and female adults to the same compound at the same concentration, and we deleted sentence “The relative EAG response ratios of males to all concentrations of (Z)-3-hexen-1-ol, β-ocimene, and 3-carene were always higher than those of females, and the relative EAG response values of males to all concentrations of α-phellandrene were always lower than those of females.. 

Point 50: L275 LSD test alone is not acceptable; P values must receive a Bonferroni correction or equivalent.

Response 50: Thank you for your correction, we have changed LSD test to Tukey’s test with Bonferroni correction.

Point 51: L280 Change “on” to “with”

Response 51: Thank you for your correction, we have corrected this.

Point 52: L280-283 Delete sentence.  It largely duplicates the next sentence and the information on the percentages is evident in the figure.

Response 52: Thank you for your correction, we have corrected this.

Point 53:L285 “were  all significantly higher than that of the control group”  to  “was significantly higher than those selecting controls”

Response 53: Thank you for your correction, we have corrected this.

Point 54: L286-287 Delete sentence. Same explanation as above.   Change “selected” to “selecting”.

Response 54: Thank you for your correction, we have corrected this.

Point 55: L300 Delete comma.

Response 55: Thank you for your correction, we have corrected this.

Point 56: L302 Change “of” to “on” 

Response 56: Thank you for your correction, we have corrected this.

Point 57: L310 Delete “to choose”

Response 57: Thank you for your correction, we have corrected this.

Point 58:L312 Change “selectivity” to “choice”

Response 58: Thank you for your correction, we have corrected this.

Point 59: L322 Delete “the”

Response 59: Done.

Point 60: L323 Change “periods” to “plants”

Response 60: Done.

Point 61: L328  Change “highest” to “most abundant”

Response 61: Done.

Point 62:L334 Replace “for that” with “and”

Response 62: Done.

Point 63:L343 Change “is hig” to “was particularly strong”

Response 63: Done.

Point 64: L349-351 Delete sentence

Response 64: Thank you for your correction, we have corrected this.

Point 65:L366 Insert “may” before “play”

Response 65: Done

Point 66: L371-378 Delete sentences.

Response 66: Thank you for your correction, we have corrected this.

Point 67: L379-381  Delete sentence.

Response 67: Thank you for your correction, we have corrected this.

Point 68: L388 Change “provides” to “should provide”

Response 68: Done.